# Graph Coloring via Learning and Metric-Guided Independent Set Extraction

## Abstract

Recent advances in Graph Neural Networks (GNNs) have enabled learning-based approaches for a wide range of combinatorial optimization problems. In this paper, we address one such classical and challenging problem: graph coloring. We introduce an algorithmic framework for graph coloring that constructs coloring sequentially by repeatedly extracting large independent sets using GNNs. The choice of independent sets for coloring is further guided by metrics which favor use of minimum number of colors. To improve scalability, we further enhance this framework with a value-aware GNN that operates on reduced graphs, learning effective independent set extraction directly on compact representations before lifting solutions to the original graph. We further illustrate the versatility of the proposed framework by applying it to quantum circuit depth optimization which is formulated as a mixed graph coloring problem. The presented approach is evaluated on standard DIMACS benchmark graphs and citation network datasets. Experimental results show that the proposed algorithm achieves competitive performance on standard benchmark instances and compares favorably with several traditional and learning-based graph coloring solvers. The results for benchmark quantum circuit instances are also encouraging.

## 1 Introduction

The graph coloring problem (GCP) is one of the fundamental problems in graph theory with wide-ranging applications in computer science, operations research, and artificial intelligence. Phrased as an optimization problem, the goal is to assign colors to the vertices of a graph $G = (V, E)$ such that no two adjacent vertices should share the same color while minimizing the total number of colors used. The minimum number of colors required to achieve such a coloring is known as the chromatic number of the graph, denoted by $\chi(G)$. Determining $\chi(G)$ is an NP-hard problem and even deciding whether a given graph is $k$-colorable is NP-complete for $k \geq 3$.

Graph coloring has been an active field of research as it plays a crucial role in many practical applications including register allocation in compilers (Chaitin et al., 1981), frequency assignment in wireless networks (Hale, 1980), examination and task scheduling (Leighton, 1979), VLSI circuit design (Sherwani, 1998), resource allocation (Cai et al., 2015) and map coloring. Over the decades, a wide range of algorithmic strategies have been proposed for solving GCP ranging from exact methods to heuristics, meta-heuristics and learning-based approaches. Exact methods include backtracking algorithms (Bender & Wilf, 1985), branch-and-bound algorithms (Méndez-Díaz & Zabala, 2006), integer linear programming (ILP) formulation (Méndez-Díaz & Zabala, 2008) etc. Then we have heuristic methods which provide efficient, often greedy procedures to generate feasible colorings, though without guarantees of optimality. The simplest among them is the greedy coloring algorithm, which processes vertices in a specific order and assigns to each vertex the smallest available color not used by its neighbors. The quality of the resulting coloring heavily depends on the vertex ordering; thus, variants such as Largest Degree ordering (LD), Smallest Last ordering (SL) and Saturation degree ordering (DSATUR) (Brélaz, 1979) have been proposed to improve performance. Another heuristic graph coloring approach is the Recursive Largest First (RLF) algorithm (Leighton, 1979) that iteratively selects large independent sets to form color classes. The algorithm exhibits an average-case time complexity of $\mathcal{O}(n^2)$ and serves as a foundation for more advanced greedy coloring schemes and benchmarks. Local search

algorithms are also widely used in GCP. One such method is TabuCol (Hertz & de Werra, 1987) which is one of the earliest and most influential tabu search algorithms developed for the graph coloring problem. This method finds a valid $k$-coloring of a graph by starting from an initial (possibly invalid) coloring and iteratively improving it through local search. In addition to these, numerous population-based evolutionary and metaheuristic approaches have also been proposed for GCP. Evolutionary algorithms exploit populations of candidate solutions and genetic operators to explore the search space effectively (Galinier & Hao, 1999), while metaheuristics such as simulated annealing, ant colony optimization and particle swarm optimization have been employed to balance exploration and exploitation in the search for high-quality colorings (Johnson et al., 1991; Costa & Hertz, 1997; Agrawal & Agrawal, 2015). These approaches have demonstrated competitive performance, particularly on large and computationally challenging graph instances.

Before GNNs were introduced, neural approaches for combinatorial optimization (CO) primarily relied on sequence-based architectures such as pointer networks (Vinyals et al., 2015). Bello et al. (2016) combined these networks with reinforcement learning (RL) to learn solution policies for several routing and optimization tasks. To better exploit the graph structure, Khalil et al. (2017) introduced a learning framework based on structure2vec graph embeddings and RL for solving problems such as Maximum Cut, Minimum Vertex Cover, and TSP. This demonstrated the value of graph-structured representations, paving the way for GNN-based combinatorial optimization. Feeding graphs to neural networks is challenging due to their complex topological structures, arbitrary size, non-Euclidean structure, no fixed ordering (isomorphism) etc. GNNs can effectively deal with all these challenges and depict excellent application in graph related problems. Recent works have successfully applied GNNs to a wide range of combinatorial optimization problems including Maximum Independent Set, GCP, routing and scheduling etc. often combining learned representations with search, RL or differentiable optimization techniques. (Schuetz et al., 2022a; Sun & Yang, 2023; Li et al., 2018b; Zhang et al., 2024b; Kool et al., 2018; Drori et al., 2020)

For the GCP problem, Lemos et al. (2019) formulates the problem as a binary classification task that is, given a graph $\mathcal{G}$ and a number $k$, is $\mathcal{G}$ $k$-colorable. They proposed GNN-GCP, a framework based on message-passing graph neural networks that jointly learn vertex and color embeddings. These learned vertex embeddings are then fed into an MLP which outputs a logit probability corresponding to the model's prediction of the answer to the decision problem. Since the ground truth of GCP is not unique, several works have also used unsupervised learning to train GNNs for solving GCP. These includes work by Schuetz et al. (2022b) in which they introduce physics-inspired GNNs. They design a specific loss function based on the Potts model in physics to obtain colorings with minimum number of conflicts. We obtain a conflict when there exist a pair of vertices which are adjacent and have received same color. Due to message-passing mechanism of GNNs, the embeddings obtained for adjacent nodes tends to be similar (Li et al., 2018a). However, Wang et al. (2024) argued that for problems like graph coloring the embeddings learned by GNN for adjacent nodes should be dissimilar and significantly apart in embedding space then only they can be labeled with different colors. They introduce GNN-NU, a GNN based on negative message passing. Li et al. (2022b) designs Graph Discrimination Network (GDN) which gives significant improvements in results compared to those obtained using standard GNNs like GCN and Sage in unsupervised setting. GNNs have also been used to design ordering heuristics for coloring (Langedal & Manne, 2025) and guide traditional algorithms for coloring (Zheng et al., 2025).

One approach to solving GCP is to begin with an initial $k$-coloring and iteratively reduce color conflicts through local improvements. Alternatively, a conflict-free coloring can be constructed by sequentially identifying independent sets and assigning each set to a distinct color class. We were interested to explore the alternate approach for building GCP solver. This approach can be beneficial in cases when obtaining a valid conflict-free coloring is the primary objective as conflict-reduction methods typically require multiple runs with different values of $k$ before a zero-conflict solution is achieved. Also, coloring via independent set extraction can be useful in certain applications where we have added constraints on the color classes obtained.

One of these applications is the problem of register allocation in compilers (Chaitin et al., 1981) which involves assigning program variables to a limited number of CPU registers such that program execution gets efficient and memory accesses is minimized. This problem can be formulated as GCP by constructing an interference graph in which each node represents a program variable and an edge connects two nodes if the

corresponding variables are live at the same time (i.e. they cannot share a register). Then assigning registers corresponds to coloring the graph such that adjacent vertices(variables) receive different colors (registers). The minimum number of colors required gives the minimum number of registers needed such that each variable gets assigned a register without any conflicts. In practice, the number of registers in compiler is fixed and usually smaller than the graph's chromatic number. In such cases, vertices assigned to colors beyond the register limit have to be spilled to memory. This makes it important not just to minimize the total number of colors but also to prioritize assigning registers to as large number of variables as possible early in the coloring process. Thus, we require a coloring method that outputs a coloring with $C_1, C_2, ..., C_k$ classes such that $|C_1| \geq |C_2| \geq \ldots \geq |C_k|$. Thus, coloring via sequential extracting independent sets could easily take care of this added requirement compared to a method that only tries to minimize colors.

In this work, we introduce Guided Beam Search (GBS), a GNN-guided framework for graph coloring that exploits learned graph representations to efficiently construct graph colorings through iterative independent set extraction. Further, we construct efficient metrics that can effectively guide the coloring process and prioritize choosing independent sets that are large enough and could simultaneously help minimize colors used. As real-world graph datasets continue to grow in size and complexity, scalability has become an increasingly important requirement for graph coloring solvers. Thus, to reduce computational overhead one may desire to reduce the size of input graph and solve the problem on a smaller reduced instance. However, reducing a graph may alter its structural characteristics, potentially affecting solution quality. Consequently, there is a need for solvers that are aware of the applied reductions and can effectively leverage the reduced graph representation to produce solutions that, when lifted back to the original graph, remain optimal or near-optimal. To this end, we design a value-aware GNN that can operate on reduced graph and extract effective independent sets with respect to the original graph. This model is subsequently integrated with GBS to construct colorings for the original graph through its reduced instance. We evaluate GBS and Value-aware GBS on DIMACS benchmark graphs and citation network datasets. The results demonstrate that our method achieves improved solution quality compared to the existing ML-based solvers while being competitive with leading non-ML methods. Furthermore, Value-aware GBS improves computational efficiency and scalability while preserving solution quality to a large extent. We also demonstrate how GBS can contribute in real world applications by using it for quantum circuit depth optimization. This is achieved by formulating it as a mixed graph coloring problem.

**Key Contributions.**

- We introduce Guided Beam Search (GBS), a GNN-guided framework for graph coloring that exploits learned graph representations to efficiently construct graph colorings through iterative independent set extraction. The coloring process is further guided by carefully designed metrics that prioritize choosing independent sets such that the coloring obtained uses minimum number of colors.

- We investigate the integration of graph reduction techniques within the GBS framework to improve scalability on large graph coloring instances. To this end, we introduce a value-aware GNN that operates on reduced graph representations and extracts independent sets for the original graph. Our results indicate that the proposed reduction-integrated approach improves computational efficiency and scalability while preserving solution quality to a large extent.

- We perform extensive experimental evaluation on DIMACS benchmark graphs and citation network datasets, comparing our approach against traditional heuristics, metaheuristic algorithms and recent ML-based solvers from the literature. The results demonstrate that our method achieves improved solution quality compared to the existing ML-based solvers while being competitive with leading non-ML methods.

- As a real-world application of the proposed framework, we demonstrate how GBS can be employed for quantum circuit depth optimization through a mixed graph coloring formulation. Experimental results on this task highlight the effectiveness of GBS and illustrate its potential for solving practical graph-structured optimization problems.

## 2 Preliminaries

### 2.1 Graph coloring problem (GCP)

Given a simple, undirected graph $G = (V, E)$ with vertex set $V = \{1, 2, 3 \ldots, n\}$ and edge set $E = \{(i, j) : i, j \in V\}$, a proper $k$-**coloring** is a function $f : V \rightarrow \{1, 2, \ldots, k\}$ such that

$$\forall \, (i, j) \in E, \; f(i) \neq f(j)$$

Graph coloring problem can be formulated in two ways. In the decision version, given a graph $G = (V, E)$ and an integer $k$, we ask whether $G$ has a proper $k$-coloring or not. Moreover, in the optimization version we find the minimum $k$ such that $G$ has a proper $k$-coloring. This $k$ is known as the chromatic number of $G$ denoted by $\chi(G)$. A graph $G$ with $\chi(G) = k$ is called $k$-chromatic and if $\chi(G) \leq k$ then $G$ is said to be $k$-colorable.

A proper $k$-coloring partitions the vertex set $V$ into $k$ independent sets called color classes (Diestel, 2017). An independent set $S$ is a subset of vertices such that no two vertices in $S$ are adjacent. $S$ is said to be **maximal** if its size cannot be increased by adding any additional vertex while preserving independence constraint. Moreover, a Maximum Independent Set **(MIS)** is an independent set of maximum cardinality among all independent sets of the graph. The cardinality of the maximum independent set in a graph $G$ is given by its independence number denoted by $\alpha(G)$.

### 2.2 Graph neural networks (GNNs)

Graphs are a natural representation for data where entities interact through complex relationships like social networks, molecular structures, citation graphs, communication networks etc. Traditional machine learning models, which typically assume vectorized inputs often struggle to process such irregular, non-Euclidean structures. This motivated the development of Graph Neural Networks (GNNs), a class of neural architectures designed specifically to learn from graph-structured data. GNNs offer a scheme to generate node representations using message-passing mechanism over the nodes and edges of a graph. The node embeddings learned incorporate the topology of the graph and are permutation invariant (i.e. node ordering does not affect output). Given a graph $G = (V, E)$ on $n$ vertices with node features $\mathbf{X} \in \mathbb{R}^{d \times n}$, a GNN updates each node's representation $h_v$ by aggregating information from its neighbors $\mathcal{N}(v) = \{u \in V : (u, v) \in E\}$. In its general form, the node embedding at layer $l$ is computed as

$$h_v^{(l)} = \text{UPDATE}^{(l)} \left( h_v^{(l-1)}, AGG^{(l)} \right)$$

$$AGG^{(l)} = \text{AGGREGATE}^{(l)} \left( \{h_u^{(l-1)} : u \in \mathcal{N}(v)\} \right)$$

where $AGGREGATE(.)$ combines messages from neighbors (e.g., sum, mean, max) and $UPDATE(.)$ combines the aggregated message with the current node state. For $l = 0$, the initial representations $h_v^0 \in \mathbb{R}^d$ are usually derived from the node's labels or given input features of dimensionality $d$. After $L$ layers of message passing, the final node embedding $h^{(L)}$ captures the local neighborhood structure upto $L$ hops. Several GNN architectures have been proposed in the literature including graph convolutional networks (GCNs), graph attention networks (GATs) and GraphSAGE (Kipf, 2016; Velickovic et al., 2017; Hamilton et al., 2017). These models mainly differ in the way information from neighboring nodes is aggregated and propagated. For instance, GCNs update node representations by aggregating normalized neighborhood features, GraphSAGE uses learnable aggregation functions over sampled neighborhoods and GATs assign different importance to neighboring nodes through attention mechanisms. These learned node embeddings can then be used for prediction tasks such as node classification. To optimize predictive power these parametrized node embeddings $h^{(L)}(\theta)$ are fed into a problem-specific loss function to optimize the weight parameters of the network.

### 2.3 GCN based MIS Extraction

A key component of our graph coloring framework is the extraction of large independent sets. To this end, we adopt the learning-based independent set generation technique proposed by Li et al. (2018b) where they

find maximum independent set (MIS) for a graph $G = (V, E)$ by estimating the likelihood for each vertex $v \in V$ of being present in MIS. Given a graph $G = (V, E)$, the goal was to produce a binary labeling for each vertex in $G$ such that label 1 indicates that a vertex is in the independent set and label 0 indicates that it's not. They learn a network $f$ that takes the graph $G$ on $N$ vertices as input and outputs $f(G)$ which is a probability map on vertices $p : V \to [0, 1]^N$ that indicates how likely each vertex is to belong to the MIS. A naive application of graph-based prediction can be insufficient due to the existence of multiple equally optimal solutions for a given graph. Since individual vertices may appear in different optimal solutions, the network may learn an averaged representation of these solutions, resulting in ambiguous and uninformative likelihood maps. Furthermore, directly converting the predicted probability map $f(G)$ into a binary assignment through simple rounding can violate the independence constraints of the MIS problem.

To overcome these challenges, the network is trained to generate multiple probability maps $\langle f^1(G_i, \theta), f^2(G_i, \theta), \ldots, f^M(G_i, \theta) \rangle$ simultaneously. Let the training dataset be denoted by $\mathcal{D} = \{(G_i, \mathbf{l}_{i_1}, \mathbf{l}_{i_2}, \mathbf{l}_{i_3}, \ldots \mathbf{l}_{i_K})\}$ where $G_i$ represents an input graph and $\mathbf{l}_{i_1}, \mathbf{l}_{i_2}, \mathbf{l}_{i_3}, \ldots \mathbf{l}_{i_K} \in \{0, 1\}^N$ denotes multiple optimal binary labelings of its vertices. A label value of 1 indicates that the corresponding vertex belongs to the MIS while 0 indicates otherwise. To learn these labelings, a multi-layer Graph Convolutional Network (GCN) Kipf (2016); Defferrard et al. (2016) is used with the following layer-wise propagation rule:

$$\mathbf{H}^{(l+1)} = \sigma \left( \mathbf{H}^{(l)} \Theta_0^{(l)} + \mathbf{D}^{-\frac{1}{2}} \mathbf{A} \mathbf{D}^{-\frac{1}{2}} \mathbf{H}^{(l)} \Theta_1^{(l)} \right)$$

Here, $\mathbf{H}^l \in \mathbb{R}^{N \times C^l}$ denotes the feature representation at layer $l$ and $C_l$ is the number of feature channels, $\Theta_0^{(l)}$ and $\Theta_1^{(l)}$ are layer-specific learnable weight matrices, $\sigma(\cdot)$ denotes the ReLU activation function, $\mathbf{A} \in \mathbb{R}^{N \times N}$ is the adjacency matrix of the graph and $\mathbf{D}$ is the degree matrix which is a diagonal matrix where $D(i, i) = \sum_j A(j, i)$. For the final layer, a sigmoid activation is applied to obtain vertex-wise probabilities. Also, the input layer $H^0$ is initialized as all ones. The network is trained using the hindsight loss Li et al. (2018c); Guzman-Rivera et al. (2012) which is defined as

$$\mathcal{L}(\mathcal{D}, \boldsymbol{\theta}) = \sum_i \min_m \ell(l_i, \ f^m(G_i; \boldsymbol{\theta}))$$

$$\ell(\boldsymbol{l}_i, f(G_i; \boldsymbol{\theta})) = -\sum_{j=1}^N \left[ \boldsymbol{l}_{ij} \log \left( f_j(G_i; \boldsymbol{\theta}) \right) + (1 - \boldsymbol{l}_{ij}) \log \left( 1 - f_j(G_i; \boldsymbol{\theta}) \right) \right]$$

where $\ell(\boldsymbol{l}_i, f^m(G_i; \boldsymbol{\theta})$ denotes the binary cross entropy loss calculated between the ground-truth label $\boldsymbol{l_i}$ for graph $G_i$ and the predicted probability map $f(G_i; \boldsymbol{\theta})$. Here, $l_{ij}$ is the ground-truth label of vertex $j$ and $f_j(G_i; \theta)$ is the predicted probability for that vertex. This objective ensures that for each training instance, only the output map with the smallest loss contributes to the parameter update. Consequently, different output heads are encouraged to learn different classes of solutions leading to a diverse set of candidate predictions. During training, the multiple valid MIS solutions available $\{\mathbf{l}_{i_1}, \mathbf{l}_{i_2}, \mathbf{l}_{i_3}, \ldots \mathbf{l}_{i_K}\}$ are cyclically sampled across epochs, allowing the network to observe different optimal configurations while retaining the diversity-promoting behavior of the hindsight loss. This formulation allows the model to capture multiple modes of the solution space rather than forcing all predictions into a single representation.

Next, the multiple probability maps generated by the network are used to generate MIS through a guided tree search procedure. Rather than constructing a single solution greedily, the search algorithm maintains a queue of partially labeled graph states. Each state represents an incomplete MIS assignment. In each step a state is chosen randomly to expand. During expansion the $M$ probability maps learned $\langle f^1(G_i, \theta), f^2(G_i, \theta), \ldots, f^M(G_i, \theta) \rangle$ are used to obtain $M$ more new complete solutions which are then added to the queue. This is similar to expanding the search tree in breadth-first manner. The algorithms used to generate MIS are given in appendix A.1.

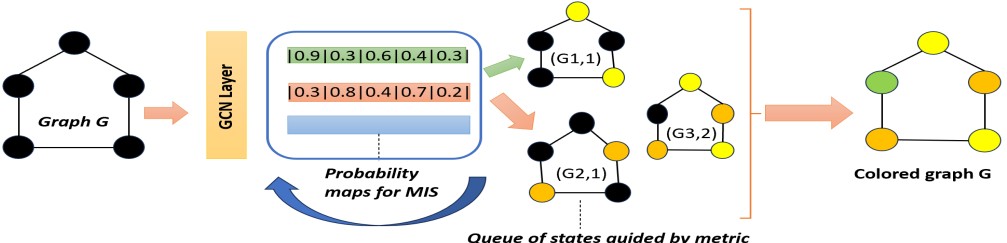

Figure 1: The figure explains the implementation of Guided Beam Search for Graph Coloring.

## 3 Guided Beam Search

We know that a proper $k$-coloring for a graph $G = (V, E)$ partitions its vertex set $V$ into independent sets (IS) called color classes. To reduce the number of colors used, it is intuitive to look for independent sets that are maximal and have large sizes so that more number of vertices can be assigned a single color class. A naive implementation could be to extract an independent set using the MIS extraction procedure discussed above, label the vertices present in MIS as one color class and repeat the procedure for the remaining unlabelled graph until all the vertices are completely labelled. This will always return a proper coloring but this approach does not optimize on the number of colors used. Also, by choosing an MIS at random out of the bunch of MIS's generated we may loose upon efficient independent sets that can help us optimize the number of colors used. For instance, out of the collection of MIS returned by MIS extraction procedure, all of them have same size yet an independent set $A$ may be more promising than independent set $B$ if for say the subgraph $G_A$ obtained after removing $A$ from graph $G$ is less dense compared to subgraph $G_B$ obtained after removing $B$. This is so because each edge corresponds to an additional coloring constraint and less density ensures fewer coloring constraint. Thus, $G_A$ can be colored further using less number of colors.

Building upon the MIS extraction procedure, we introduce the Guided Beam Search (GBS) algorithm, a search-based framework that efficiently navigates through the exponentially large state space consisting of graph states of form $(G, c)$ where graph $G$ is partially colored and $c$ denotes the number of colors used so far by labeled vertices of $G$. This is achieved by maintaining a queue of fixed size consisting of most promising graph states. At each step, for the given partially labeled graph, the corresponding graph consisting of only unlabeled vertices is fed to network for extracting multiple MIS candidates for the unlabeled subgraph. For every candidate MIS, a new graph state is created by labeling the vertices present in MIS as one color class. These new states are then evaluated using a predefined metric function that estimates the potential of each state toward an efficient coloring. Only the most promising states as determined by this metric and constrained by the beam width are retained for further exploration. This process continues until all vertices are assigned a color. Algorithm 1 outlines the guided beam search approach.

The algorithm predominantly uses two functions, $find\_multiple\_MIS(.)$ and $update\_queue(.)$. The $find\_multiple\_MIS(.)$ function outputs multiple MIS sets for the graph using MIS extraction approach discussed above. On the other hand the $update\_queue(.)$ function sorts queue($Q$) based on a pre-defined metric and keep at most $c$ graph states ($c$ is queue size) with minimum value for the metric. Note that since the network $f$ is trained to output maximum independent set for a graph $G$, the independent sets obtained will be of large size and will also be maximal. As discussed earlier, this added property of color classes can be beneficial for a lot of applications. The efficiency of the algorithm significantly depends upon the MIS extracted and choice of metric used to evaluate and rank intermediate graph states. These also influence the exploration of solution space for finding efficient colorings. Next, we discuss the different metrics that we designed for use in the algorithm. For a given graph $G = (V, E)$, a graph state $(G'(V'_l, V'_{un}, E), c)$ denotes that graph $G$ is partially colored with $V'_l \subseteq V$ denoting the subset of labeled vertices which are labeled using $c$ colors and $V'_{un}$ denoting the set of unlabeled vertices. Also, we denote the subgraph containing

---

**Algorithm 1** Guided Beam Search for Vertex Coloring

---

**Input:** Graph $G = (V, E)$, queue capacity $c$
**Output:** Coloring with minimum number of colors
Initialize queue $Q$ with capacity $c$
Initialize $min\_colors \leftarrow \infty$
Enqueue $(G, 0)$ into $Q$
**while** $Q$ is not empty **do**
   $(G_{current}, num\_colors) \leftarrow$ Dequeue from $Q$
   **if** $G_{current}$ has no vertices **then**
      $min\_colors \leftarrow \min(min\_colors, num\_colors)$
      continue
   **end if**
   $MIS\_list \leftarrow$ find\_multiple\_MIS$(G_{current})$
   **for** each $MIS$ in $MIS\_list$ **do**
      $G_{new} \leftarrow G_{current}$ with $MIS$ removed
      Enqueue $(G_{new}, num\_colors + 1)$ into $Q$
   **end for**
   update\_queue$(Q)$
**end while**
**return** $min\_colors$

---

only unlabeled vertices by $G[V'_{un}]$. Here, the edge set of $G[V'_{un}]$ is the edges between vertices in $V'_{un}$. For simplicity, the graph state is denoted by $(G', c)$.

- **Edge density:** Each edge $(u, v)$ corresponds to a constraint that vertices $u$ and $v$ cannot share the same color. Thus, graph states with minimal edge density would impose fewer coloring constraints. For a graph state $(G'c)$, the metric is defined as

$$M_{ed}((G'c)) = \frac{\text{no. of edges in } G[V'_{un}]}{\text{Total no. of possible edges in } G[V'_{un}]} = \frac{|E(G[V'_{un}])|}{\binom{|V'_{un}|}{2}}$$

- **Edge density with colors used:** While edge density is a useful measure, relying on it alone can lead to suboptimal decisions as it ignores the number of colors used so far. For example, consider two graph states $(G_1, c_1)$ and $(G_2, c_2)$ such that $M_{ed}(G_1) < M_{ed}(G_2)$ and $c_1 > c_2$. Then $G_1$ appears more favorable under the edge density metric, but it could have reached its current form only after multiple MIS extractions, using many colors. In contrast, $G_2$ might still be in early stages of coloring and may offer better long-term potential. To account for this trade-off, we propose the combined metric $M_{edc}$ for graph state $(G', c)$ given by

$$M_{edc}((G', c)) = M_{ed}((G', c)) \times \frac{c}{\Delta(G) + 1}$$

where $\Delta(G)$ denotes maximum degree of input graph $G$. This metric favors graph states which are sparse while simultaneously using minimum number of colors.

- **Average closeness centrality:** Closeness centrality is a graph-theoretic measure that quantifies how close a given node is to all other nodes in the graph, based on the lengths of the shortest paths. For a connected graph $G = (V, E)$, the closeness centrality of a node $v \in V$ is defined as

$$C(v) = \frac{n-1}{\sum_{u \in V, u \neq v} d(v, u)}$$

where $n = |V|$ is the number of vertices and $d(v, u)$ is the shortest path distance between nodes $v$ and $u$. The average closeness centrality of the graph is then defined as the mean value of $C(v)$ over

all vertices. Thus, for a graph state $(G', c)$ we find average closeness centrality for the unlabeled subgraph $G[V'_{un}]$. The metric is defined as

$$M_{acc}((G'c)) = \frac{1}{n} \sum_{v \in V'_{un}} C(v)$$

In the context of graph coloring, this metric can serve as a quantifier for the complexity of the remaining graph structure. A subgraph with high average closeness centrality tends to have more interconnected and centrally positioned nodes, which may lead to more coloring conflicts due to tighter clustering. On the other hand, a graph with lower average closeness centrality may be more tree-like or loosely connected, potentially allowing easier decomposition into independent sets.

- **Color efficiency:** To minimize number of colors used one can also look for maximizing the number of vertices per colors class. This can be done using color efficiency metric which for a graph state $(G', c)$ is defined as

$$M_{ce}((G', c)) = \frac{|V'_l|}{c}$$

A higher color efficiency indicates that more vertices are being colored with fewer colors, which is a desirable property in graph coloring.

- **Progress rate:** This metric keeps a track of number of vertices that have already been colored. For coloring graph $G$ with intermediate state $(G', c)$ it is defined as

$$M_{pr}((G'c)) = \frac{|V'_l|}{V(G)|}$$

## 4 Weighted Beam Search

In this section, we aim to enhance guided beam search by employing a weighted combination of multiple metrics with the weights predicted by a neural network trained to adapt to the structural properties of each input graph. The single metric score is replaced with a weighted metric $M_w$ which is weighted sum of four metrics namely edge density ($M_{ed}$), average degree ($M_{avg}$), color efficiency ($M_{ce}$) and progress rate ($M_{pr}$).

$$M_w = w_1.M_{ed} + w_2.M_{avg} + w_3.M_{ce} + w_4.M_{pr}$$

Here, metric average degree ($M_{avg}$) computes average degree of unlabeled subgraph $G[V'_{un}]$ in intermediate state $(G', c)$ and normalizes by max degree of initial graph $G$. To determine effective weights for this weighted metric for each graph instance, we developed a two-stage learning pipeline which included training data generation and neural network training.

For generating training data we chose a representative subset of training graphs and performed random search over possible weight combinations to identify those that yield the best coloring results (i.e., minimum number of colors used). We obtain a training dataset $\mathcal{D} = \{(\mathbf{v}(G_i), \boldsymbol{w}_i)\}$ in which corresponding to every training graph $G_i$, a vector $\mathbf{v}(G_i) \in \mathbb{R}^{10}$ is computed to capture its structural characteristics. Specifically, the vector comprises ten graph descriptors: the number of vertices, the number of edges, graph density, average degree, maximum degree, minimum degree, standard deviation of vertex degrees, graph diameter, average shortest-path length, and average clustering coefficient. Each graph $G_i$ is labeled with the optimal combination of weights $\boldsymbol{w}_i \in [0, 1]^4$ obtained for it. Using this dataset $\mathcal{D}$, a lightweight feed-forward neural network referred to as the $Weight\_Predictor(.)$ is trained that takes as input a feature vector $\mathbf{v}(G) \in \mathbb{R}^{10}$ as described above and outputs optimal weight vector $\mathbf{w} \in [0, 1]^4$ for graph $G$ that can be used while finding optimal coloring for graph $G$ using GBS which now uses weighted metric $M_w$ inside $update\_queue(.)$ function. This adaptive weighting mechanism allows the GBS algorithm to dynamically modify its metric prioritization to the underlying structure of the input graph, resulting in improved performance over using a fixed metric across all graph instances.

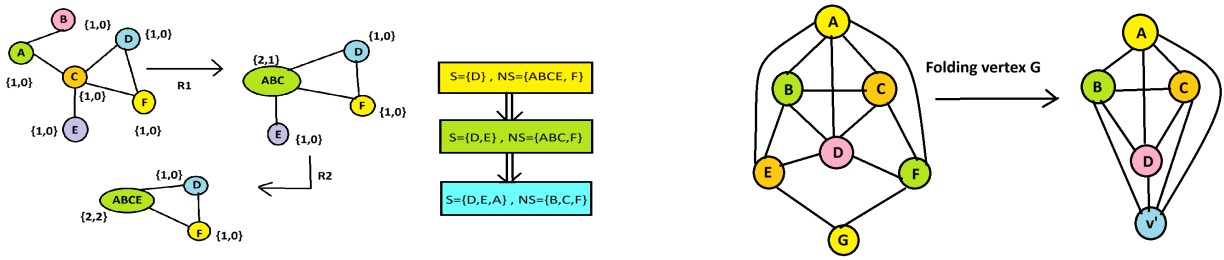

Figure 2: In first figure, a graph is reduced by first folding vertex $A$ followed by folding vertex $E$. $S$ and $NS$ denotes sets of selected and non-selected nodes respectively. It is shown how sets $S$ and $NS$ gets updated as vertices are unfolded back. In second figure, original graph uses 4 colors whereas the reduced graph uses 5 colors in optimal coloring.

## 5  Value-aware GNN for graph reductions

Real-world graphs such as citation networks, social networks and communication systems etc. often consist of millions of nodes and edges. Applying graph coloring algorithms directly to such large-scale graphs can be computationally challenging. To address this, we explore graph reduction techniques and how they can be incorporated in the framework discussed above.

**Definition 5.1.** A *graph reduction* is a mapping $R : \mathcal{G} \to \mathcal{G}$ over the set of graphs $\mathcal{G}$ that maps a graph $G = (V, E)$ to a smaller graph $G' = (V', E')$ (known as the reduced graph) such that $|V'| < |V|$ and/or $|E'| < |E|$ while preserving certain structural or problem-specific properties of $G$.

For coloring we explore two reduction techniques namely vertex folding and pendant folding which are also mentioned in paper by Li et al. (2018b).

- **Vertex folding:** This technique focus on vertices of degree 2 having non-adjacent neighbors. If $deg(v) = 2$ and $N_v(G) = \{u, w\}$ such that $uw \notin E(G)$ then $u, v, w$ are deleted and representative vertex $v'$ is added. All neighbors of $u$ and $w$ are joined to $v'$.

- **Pendant folding:** This technique targets the set of leaf nodes $L(G)$ (degree 1 vertices). If $v \in L(G)$ such that $N_v(G) = u$ then $u$ and $v$ are deleted and representative vertex $v'$ is added. All the neighbors of $u$ are joined to $v'$.

Such reductions can be used as pre-processing steps for a graph $G$ and reduced graph $G'$ can be given as input to obtain graph coloring for $G$. Note that in cases where we obtain $min\_colors = 1$ using GBS for the reduced graph, we need to increase color count by 1 as the original graph will require 2 colors when the vertices will be unfolded.

For a given reduction function, size of maximum independent set need not be preserved. For example, for graph $G$ in figure 2 the reduced graph has an MIS of size 1. If either of sets $\{D\}$ or $\{F\}$ is chosen as MIS for reduced graph then MIS size will be 1 for original graph also. Instead, if node $ABCE$ is chosen then unfolding vertices will generate $\{A, E\}$ or $\{B, C\}$ as MIS which are of size 2. However, original graph has an MIS of size 3. Also in some cases, the reduced graph obtained may have chromatic number $\chi(G') > \chi(G)$ (see figure 2) This happens when the reduced graph becomes structurally complex and dense due to reduction.

Thus, it becomes necessary to incorporate reduction information in the reduced graph such that vertices in MIS of reduced graph are chosen wisely and we get independent set of large sizes as we unfold to obtain MIS for original graph. This can be done by labeling each node with select and nonselect values which will measure its contribution to original MIS if it is being chosen or not. This value-tracking mechanism will help in preserving cardinality information throughout the reduction process. Select and nonselect values for vertex folding a vertex $v$ with non-adjacent neighbors $u$ and $w$ are governed by equations (1) and (2) and for pendant folding vertex $v$ ( degree 1) with its neighbor $u$ by equations (3) and (4). These values are initialized

to 1 and 0 respectively for all original (unmerged) vertices and then updated systematically based on graph reduction technique used (see figure 2).

$$\text{select}(v') = \text{select}(u) + \text{select}(w) + \text{nonselect}(v) \tag{1}$$

$$\text{nonselect}(v') = \text{select}(v) + \text{nonselect}(u) + \text{nonselect}(w) \tag{2}$$

$$\text{select}(v') = \text{select}(u) + \text{nonselect}(v) \tag{3}$$

$$\text{nonselect}(v') = \text{nonselect}(u) + \text{select}(v) \tag{4}$$

Once a maximum independent set $S$ is identified in the reduced graph, the exact cardinality of the corresponding maximal independent set in the original graph can be computed using the value-aware representation:

$$\text{cardinality} = \sum_{v \in S} \text{select}(v) + \sum_{v \notin S} \text{nonselect}(v)$$

This formulation provides an efficient mechanism for estimating the quality of independent sets without reconstructing the full original graph. To incorporate this value-tracking mechanism, we learn *value_aware_multiple_MIS*(.) which similar to *find_multiple_MIS*(.) finds multiple value-aware MIS for $G$ by taking reduced graph $G'$ as input. Value-aware MIS for reduced graph $G'$ is an independent set $I'$ of $G'$ which will produce MIS for original graph after unfolding vertices. Unfolding vertices here means that for each representative vertex $v'$ we add the corresponding vertices from $G$ based on whether $v'$ belongs to $I'$ or not. Algorithm 2 explains the working of value-aware GBS approach. Compared to GBS, this algorithm works directly on the reduced value-aware graph in which each node $v$ has a tuple $(select(v), nonselect(v))$ associated with it. While generating a new graph $G_{new}$ from the current graph select and non-select values are updated according to the chosen MIS. If $v \in MIS$, then $select(v)$ is updated to 0 making sure that $v$ will not contribute any select value further as it is already being selected by the MIS. Similarly, if $v \notin MIS$ then $nonselect(v)$ is updated to 0. Also, there may arise a case when $select(v) = 0$ but $nonselect(v) > 0$, such a node is made isolated so that next time it is selected in MIS and the vertices corresponding to nonselect value are being colored. Finally, if $select(v) = 0$ and $nonselect(v) = 0$ then such a node is removed from $G_{current}$.

For effectively learning this network we modify node embeddings and loss function. Node embeddings are initialized with the value pair $[select(v), nonselect(v)]$ for each vertex $v$, instead of uniform or identity features as in standard GCN settings. The training dataset $\mathcal{D} = \{(G_i, \boldsymbol{l}_{i_1}, \boldsymbol{l}_{i_2}, \boldsymbol{l}_{i_3}, \ldots \boldsymbol{l}_{i_K})\}$ now contains reduced graph in which each vertex has a select value and non-select value associated with it. For each reduced graph, multiple labelings are generated such that each of these labelings denotes an independent set in reduced graph which upon unfolding generates a MIS in original graph. The GCN is trained to predict membership in independent sets (probability maps on vertices) that maximize the *total select value* rather than merely the number of nodes. To effectively train the GCN under this modified objective, we propose a Value-Aware Binary Cross-Entropy (VA-BCE) loss function which is used in hindsight loss rather than simple binary cross entropy. VA-BCE loss function weights the loss contribution of each node according to its selectivity:

$$\ell_{\text{VA-BCE}}(\boldsymbol{l}_i, f(G_i; \boldsymbol{\theta})) = -\frac{1}{N} \sum_{j=1}^{N} w_j \Big[ \boldsymbol{l}_{ij} \log \big( f_j(G_i; \boldsymbol{\theta}) \big) + (1 - \boldsymbol{l}_{ij}) \log \big( 1 - f_j(G_i; \boldsymbol{\theta}) \big) \Big] \tag{5}$$

where $\boldsymbol{l_i}$ is the ground-truth label for graph $G_i$, $f(G_i; \boldsymbol{\theta})$ is the predicted probability map and $N$ is the number of vertices in $G_i$. Here, $\boldsymbol{l}_{ij}$ is the ground-truth label of vertex $j$, $f_j(G_i; \boldsymbol{\theta})$ is the predicted probability for that vertex and $w_i$ is the value-aware weight defined as:

$$w_j = \begin{cases} \frac{s_j}{\bar{s}} & \text{if } \boldsymbol{l}_{ij} = 1 \\ \frac{ns_j}{\overline{ns}} & \text{if } \boldsymbol{l}_{ij} = 0 \end{cases} \tag{6}$$

where $s_j = \text{select}(v_j)$, $ns_j = \text{nonselect}(v_j)$, and $\bar{s}, \overline{ns}$ are the mean values over all nodes with the respective labels. This value-aware loss encourages the model to prioritize including nodes with high select values and

---

**Algorithm 2** Value aware coloring with Reduction $R$

---

**Input:** Graph $G = (V, E)$, queue capacity $c$, Reduction function $R$
**Output:** Coloring with minimum number of colors
Initialize queue $Q$ with capacity $c$
Initialize $min\_colors \leftarrow \infty$
Generate value-aware graph, $G_{reduced} \leftarrow R(G_{current})$
Enqueue $(G_{reduced}, 0)$ into $Q$
**while** $Q$ is not empty **do**
    $(G_{current}, num\_colors) \leftarrow$ Dequeue from $Q$
    **if** $G_{current}$ has no vertices **then**
        $min\_colors \leftarrow \min(min\_colors, num\_colors)$
        continue
    **end if**
    $MIS\_list\_red \leftarrow$ value_aware_multiple_MIS$(G_{reduced})$
    **for** each $MIS$ in $MIS\_list\_red$ **do**
        $G_{new} \leftarrow update\_graph(G_{current}, MIS)$
        Enqueue $(G_{new}, num\_colors + 1)$ into $Q$
    **end for**
    update_queue$(Q)$
**end while**
**return** $min\_colors$

---

excluding nodes with high nonselect values, leading to more informed and effective MIS predictions. By explicitly encoding node-level contribution information, this approach enhances the GCN's ability to learn the structural and value-based dynamics that arise during graph reduction. We provide empirical evidence of its effectiveness on several representative graph instances in section 7.4.

# 6 Application: Quantum Circuit Depth Optimization

In this section, we consider the practical problem of quantum circuit optimization and demonstrate how the proposed GBS framework can be applied to solve real-world graph-structured optimization problems. Quantum algorithms are typically implemented using the quantum circuit model in which computations are represented as sequences of quantum operations (known as quantum gates) acting on qubits. These circuit models are highly susceptible to errors and inefficiencies due to quantum noise and the limited capabilities of existing quantum hardware. Also, quantum computers are highly susceptible to decoherence which is qubit loss of quantum information over a period of time. Thus, optimizing quantum circuits becomes critical for enhancing computational speed and mitigating errors caused by quantum noise. Over the past years, researchers have devised innovative techniques and algorithms to optimize quantum circuits focusing on characteristics of circuits such as circuit depth, gate count, qubit count, gate fidelity etc.(Karuppasamy et al., 2025; Saeedi & Markov, 2013; Nam et al., 2018; Huang & Sun, 2022)

In this paper, we mainly focus on optimizing circuit depth. For a given quantum circuit, some operations in it can be parallelizable (can be simultaneously executed on the system) depending on the hardware machine. This concept of parallelizability leads to the notion of depth (of a circuit). A layer in a circuit is defined as the collection of gates that are parallelizable. The *depth* of a circuit corresponds to the number of consecutive layers executed during its operation. Circuits with smaller depth are generally preferable as shorter execution time reduces exposure to decoherence, limits accumulation of noise and thereby improves fidelity.

Rearranging commuting gates can help in reducing depth. For example, consider circuit $C$ in figure 3. The two gates $CNOT(q_3, q_1)$ and $CNOT(q_3, q_4)$ commute. This allows to rearrange the circuit and now $CNOT(q_3, q_4)$ can be executed with gate $R_Z(\pi/2)$ in parallel. Recent work by Lee et al. (2025) use this property of commuting circuits and model circuit depth minimizing problem as graph coloring task where quantum gates are vertices and edges capture commutation-based conflicts. However, their model is restricted

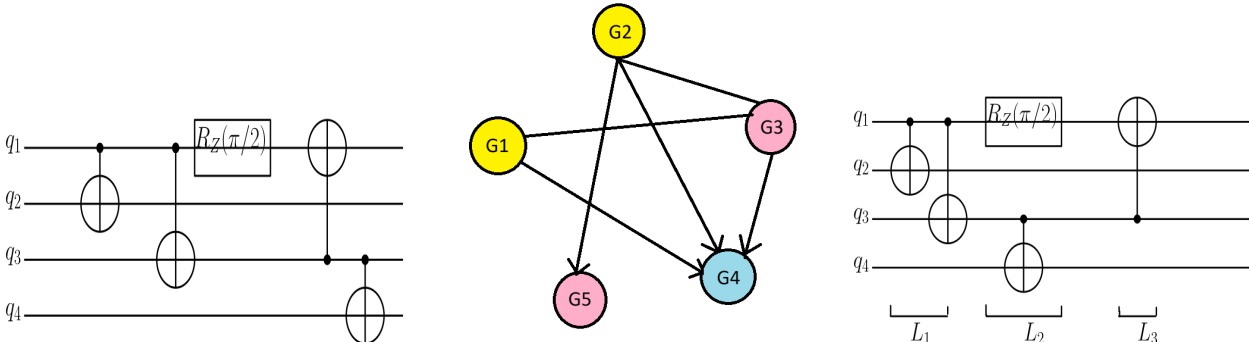

Figure 3: Sample circuit $C$ ($depth = 5$), its corresponding dependency graph $G$ and the optimized circuit $C'$ ($depth = 3$).

only for commuting circuits i.e. quantum circuit in which every pair of gates commutes. However, quantum circuits in general need not be commuting. For example, in circuit $C$ (in figure 3) $CNOT(q_1, q_3)$ and $CNOT(q_3, q_4)$ do not commute. Thus, we also require directed edges to enforce execution order due to non-commuting gate dependencies. We generalize the prior formulation given by Lee et al. (2025) for arbitrary circuits by modeling the quantum circuit as a mixed graph $G(V, E_d, E_u)$ which includes both, directed edges ($E_d$) to enforce execution order and undirected edges ($E_u$) to prevent simultaneous execution of commuting gates that act on overlapping qubits.

To perform graph coloring, we transform a quantum circuit into a *dependency graph* $G(V, E_d, E_u)$ where each node represents a quantum gate and edges are added based on dependencies. A directed edge from gate $g_1$ to $g_2$ represents a hard dependency that $g_1$ must be executed before $g_2$. This occurs when both gates act on overlapping qubits and do not commute. An undirected edge between $g_1$ and $g_2$ implies that the two gates commute but share at least one qubit. Given a dependency graph $G(V, E_d, E_u)$, we require a suitable coloring that can be converted to an execution order that satisfies the constraints imposed in a given circuit.

**Definition 6.1.** A *suitable coloring* $f : V \to \mathbb{N}$ is one that assigns color labels such that for each directed edge $(u, v) \in E_d$, $f(u) < f(v)$ which enforces the correct execution order and for each undirected edge $(u, v) \in E_u$, $f(u) \neq f(v)$ which ensures that conflicting gates are not executed in parallel.

Corresponding to the obtained coloring, gates must be arranged in the sequence of increasing color label. Gates labeled with same color can be executed in parallel and thus they together form a layer. Then the circuit depth is given by $max_{v \in V} f(v)$ which can be minimized by minimizing the number of colors used. Now, any arbitrary vertex coloring algorithm could work only for an undirected graph and could not be directly applied for finding suitable coloring for the mixed graph that we have obtained. This is where the proposed GBS framework proves particularly useful. One may observe that, a gate can be placed in the first layer of the circuit only if its indegree is 0. (ie. it does not have an incoming edge incident to it in the dependency graph). Moreover, amoung such gates one can select a subset of gates that don't have an edge between them and place them together in layer 1 for parallel execution. Thus, to increase parallelization we need to find a maximum size subset of gates such that they form an independent set in dependency graph. Since, GBS approaches the coloring problem through the extraction of independent sets, this framework can be easily used for minimizing circuit depth by reducing the number of colors required to obtain a suitable coloring for the dependency graph.

Each time we choose gates for a layer, we require that either gates have indegree 0 or their in-neighbors are already assigned a previous layer ensuring a correct execution order. This is achieved inside GBS framework by introducing additional preprocessing and postprocessing steps. Given a dependency graph $G(V, E_d, E_u)$, we extract a subgraph $G_{exec}(V', E_u')$ which only consist of vertices with indegree 0 and corresponding undirected edges between these vertices. There always exist a vertex with indegree 0 i.e. the vertex corresponding to 1st gate. Then the GBS algorithm finds a suitable independent set in $G_{exec}$. Once an independent set

is identified, graph $G$ is updated to $G'$ by removing the colored nodes and updating the indegrees of all remaining nodes. The process is repeated again for updated graph $G'$ until all nodes in $G$ are colored. The steps added above ensures that the coloring obtained will be suitable. (Refer to Algorithm 5 in appendix).

**Theorem 6.2.** *The optimized circuit $C'$ obtained from above algorithm is executable i.e. the coloring obtained is suitable.*

**Proof:** The proof is deferred in Appendix A.4

## 7 Numerical Experiments

In this section, we evaluate the proposed GBS framework against a diverse set of baseline approaches including traditional heuristics, metaheuristic algorithms and machine learning-based solvers. We further investigate the performance of the Value-aware GBS variant to assess its ability to reduce computational cost while maintaining solution quality. Finally, we demonstrate the applicability of the proposed framework to the problem of quantum circuit depth optimization by conducting experiments on a collection of benchmark quantum circuits.

### 7.1 Experimental setup

We solve GCP on DIMACS coloring dataset (Johnson & Trick) as well as on citation network datasets (Cora, Citeseer, Pubmed (Yang et al., 2016) and Ego-Facebook (Leskovec & Mcauley, 2012)) which are often used for graph-based benchmark experiments. The DIMACS dataset contains both synthetic and real-world graphs exhibiting diverse structural characteristics and varying levels of difficulty which makes them a suitable benchmark for evaluating both solution quality and computational efficiency. On the other hand the citation network datasets enable us to assess the performance of the proposed method on large, naturally occurring graphs arising from real-world applications. We consider them simply as undirected graphs without node or edge features. For details on network settings and training one can refer A.2.

For a broader evaluation, we compare GBS not only against learning-based approaches but also with several traditional heuristics based algorithms and state-of-the-art non-ML solvers such as metaheuristic-based algorithms. The heuristics based algorithms used for comparison are LF, SL, DSatur (Brélaz, 1979) and RLF (Leighton, 1979). Further, the ML based solvers used for comparison are PI-GNN (Schuetz et al., 2022b) and GNN-NU Wang et al. (2024). Metaheuristics based algorithms used for comparison are swarm optimization based methods such as CSO (Saeed et al., 2024)and CASCOL (Ge et al., 2010),TLBO (Dokeroglu & Sevinc, 2021) which uses teaching learning based optimization, (Dokeroglu et al., 2025),NERS-HEAD (Guo & Guo, 2023) and ACOQA (Kole & Pal, 2025).

### 7.2 Performance of GBS for GCP

For testing GBS, we took 71 graphs instances from DIMACS coloring dataset. Amoung these instances, chromatic number is known for 51 graphs. For these graphs, the GBS algorithm obtained correct chromatic number for 57% of graphs while for other graphs extra colors used were at most 4. We begin by comparing GBS and W-GBS with traditional heuristics such as LF, SL, DSatur (Brélaz, 1979) and RLF (Leighton, 1979). Results obtained for some of the graph instances are reported in Table 1. All of these heuristics are briefly described in appendix A.2. Both the GBS and W-GBS performs much better compared to these standard heuristics.

Next, for comparison with ML-based models we select PI-GCN, PI-SAGE (Schuetz et al., 2022b), GNN-NU (Wang et al., 2024) and GIN (Zhang et al., 2024a). Given $k$, all of these methods are trained to find a $k$-coloring with minimum number of conflicts. A conflict arises in a given $k$-coloring if there exist an edge with endpoints labeled with same color. For a fair comparison with GBS which always outputs a coloring with zero conflicts, for all these methods we report the minimum value of $k$ required to obtain a $k$-coloring with zero conflicts. While running experiments, we observed that the effectiveness of these methods is strongly influenced by their hyperparameter settings which thus needs to be fine-tuned for each instance individually to obtain high-quality results. This increases the overall training time making these methods impractical

Table 1: Coloring results and comparison with traditional heuristic algorithms. Number of colors used by optimal coloring obtained from each solver is reported.

| Graph Name | $|V|$ | $|E|$ | $\chi$ | GBS | W-GBS | RLF | DSatur | LF | SL |
|---|---|---|---|---|---|---|---|---|---|
| queen7_7 | 49 | 952 | 7 | 7 | 7 | 9 | 11 | 13 | 12 |
| queen8_8 | 64 | 728 | 9 | 10 | **9** | 11 | 12 | 13 | 14 |
| queen11_11 | 121 | 3960 | 11 | 13 | **13** | 14 | 15 | 19 | 18 |
| queen13_13 | 169 | 6656 | 13 | 15 | **15** | 16 | 17 | 22 | 20 |
| queen15_15 | 225 | 10360 | ? | 17 | **16** | 19 | 21 | 25 | 23 |
| le450_5a | 450 | 5714 | 5 | 8 | **7** | 8 | 10 | 11 | 11 |
| le450_5b | 450 | 5734 | 5 | 8 | 8 | 8 | 9 | 12 | 13 |
| DSJC125.9 | 125 | 13922 | ? | 49 | **46** | 49 | 51 | 53 | 52 |
| DSJC250.5 | 250 | 15668 | ? | 33 | **31** | 34 | 37 | 41 | 39 |
| DSJC250.9 | 250 | 27897 | ? | 79 | **76** | 83 | 92 | 93 | 96 |
| DSJC500.5 | 500 | 62624 | ? | 54 | **53** | 60 | 65 | 71 | 70 |
| DSJC1000.1 | 1000 | 49629 | ? | 25 | 24 | 24 | 27 | 29 | 30 |
| DSJC1000.5 | 1000 | 249826 | ? | 95 | **94** | 107 | 115 | 121 | 125 |
| school1 | 385 | 19095 | ? | 19 | **18** | 27 | 17 | 32 | 15 |
| flat300_20_0 | 300 | 21375 | 20 | 21 | **21** | 37 | 42 | 44 | 46 |

Table 2: Coloring results and comparison with metaheuristics based algorithms. Number of colors used by optimal coloring obtained from each solver is reported. Metric used for GBS is edge density with colors used.

| Graph Name | $|V|$ | $|E|$ | $\chi$ | GBS | CSO | CAS-COL | TLBO | NERS-HEAD | ACOQA |
|---|---|---|---|---|---|---|---|---|---|
| flat300_20_0 | 300 | 21375 | 20 | 21 | - | 30 | 20 | - | 20 |
| le450_25b | 450 | 8263 | 25 | 27 | 28 | 26 | 25 | 25 | 25 |
| DSJC125.1 | 125 | 736 | ? | 5 | 8 | 7 | 5 | 5 | 5 |
| DSJC125.5 | 125 | 3891 | ? | 19 | 26 | 24 | 17 | 17 | - |
| DSJC125.9 | 125 | 6961 | ? | 47 | 56 | 53 | 44 | 44 | 45 |
| DSJC250.1 | 250 | 3218 | ? | 10 | 13 | 12 | 8 | 8 | - |
| DSJC250.5 | 250 | 15668 | ? | 31 | 43 | 40 | 28 | 28 | - |

for large graph instances. In contrast, GBS employs the same set of parameter values across all graph instances. We discuss more on the impact of parameters in section 7.5. We also have another ML-based method GNN-GCP (Lemos et al., 2019) which frames GCP as a binary classification problem that is given $k$ it classifies whether $G$ is $k$-colorable or not. Since the model is trained purely as a classifier rather than a coloring generator, it may occasionally predict a graph to be $k$-colorable even when its chromatic number exceeds $k$. For this reason, it is not included for comparing with other models.

In comparison to the several state-of-the-art metaheuristic algorithms such as CSO Saeed et al. (2024), CASCOL Ge et al. (2010), TLBO Dokeroglu & Sevinc (2021),NERS-HEAD Guo & Guo (2023) and ACOQA Kole & Pal (2025), GBS is able to produce near-optimal colorings which uses slightly more number of colors than the best known results. These results are reported in Table 2. We implemented CASCOL following the description provided in the original paper. Nevertheless, due to the non-deterministic nature of the algorithm, there may be differences between reported results and the original results. For the remaining methods, results are taken directly from the cited works. A dash (–) indicates that no result was reported for the corresponding instance. Therefore, GBS not only enhances the performance of ML-based graph coloring approaches but also remains competitive with leading non-ML methods. Also, as discussed earlier, the coloring obtained by GBS comes with an additional property that is for coloring $C$ with color classes $(C_1, C_2, \ldots, C_k)$ we have $|C_1| \geq |C_2| \geq |C_3| \ldots \geq |C_k|$. This characteristic enhances the practical utility of GBS and makes it more suitable compared to other graph coloring solvers for a lot of applications such as register allocation, quantum circuit depth optimization etc.

| Graph Name | $\chi$ | GBS | PI-GCN | PI-SAGE | GNN-NU | GIN |
|---|---|---|---|---|---|---|
| **myciel5** | 6 | 6 | 6 | 6 | 6 | 6 |
| **myciel6** | 7 | 7 | 7 | 7 | 7 | 7 |
| **queen5__5** | 5 | 5 | 5 | 5 | 5 | 5 |
| **queen6__6** | 7 | 9 | 8 | **7** | **7** | **7** |
| **queen7__7** | 7 | **7** | 8 | **7** | **7** | **7** |
| **queen8__8** | 9 | 10 | 10 | 10 | 10 | **9** |
| **queen9__9** | 10 | **11** | 12 | **11** | **11** | **11** |
| **queen8__12** | 12 | 13 | 14 | **12** | **12** | 13 |
| **queen11__11** | 11 | 13 | 15 | 14 | 14 | **12** |
| **queen13__13** | 13 | **15** | 18 | 17 | 17 | 16 |

Table 3: Coloring results and comparison with Tabucol and GNN-based models.

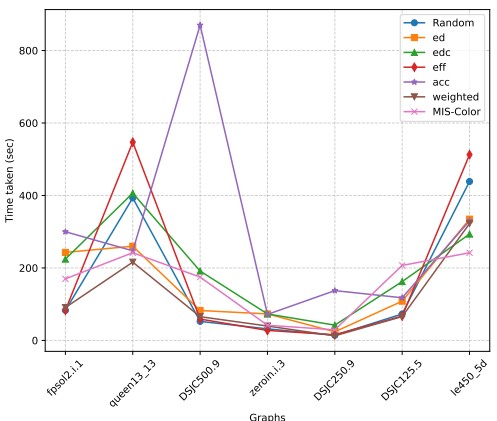

Figure 4: Comparison of execution time of GBS under different MIS-selection metrics

## 7.3 Metric analysis

In Section 3 we introduce several metrics that can be used in GBS framework to enhance its performance. Here, we compare the following metrics such as edge density (ed), edge density with colors used (edc), color efficiency (eff), average closeness centrality (acc) and weighted metric (w). To assess the necessity of the metric within GBS, we also evaluate the framework in the absence of any metric-based guidance. In the absence of metric, queue is updated by randomly picking graph states equal to the size of queue and discarding the remaining states. Also, one may simply consider sequentially extracting MIS from method by Li et al. (2018b) and assigning each independent set a distinct color class. Thus, we refer to this method as MIS-Color and include this as a baseline for comparison. Table 4 compares the performance of GBS without metric(Random) with using different metrics and also with baseline MIS-Color. For each method, the average number of colors used over 10 independent runs along with the corresponding standard deviation is reported. It is evident that selecting graph states at random or following MIS-Color strategy, both of them results in a higher average number of colors compared to the metric-guided variants. Furthermore, they exhibit a larger standard deviation, indicating less consistent performance across runs. This can be attributed to the fact that MIS-Color focuses on extracting large maximal independent sets without explicitly optimizing the number of colors used. Likewise, in the absence of a guiding strategy, randomly selected independent sets are often ineffective at reducing the total number of colors used. Thus, we require additional guidance by metrics to choose adequate independent sets that are not only large in size but will also help in reducing number of colors used in long run until the graph is completely colored. Now, comparing among metrics, it is evident that a single metric cannot perform well for all graph instances. However, we observe that weighted metric performs significantly well for most of the instances and also reports less computational cost as can be observed in figure 4. Since, these weights are uniquely assigned for each graph instance based on its structural property, the MIS selection procedure is further improved. Also, for all the metrics and most of the graph instances, the results exhibit low variance across multiple runs indicating that the proposed method is stable and robust.

## 7.4 Value aware GBS

For testing value-aware GBS we considered few instances from DIMACS dataset which showed significant reduction in size after applying vertex folding and pendant folding reductions. Also citation graphs and social network graphs like Cora,Citeseer,Pubmed and Ego-Facebook are considered to test model efficiency for large-scale graphs. Compared with applying GBS on original graph, the Value-aware GBS method applied on reduced instances could significantly reduce runtime while maintaining efficiency. The Value-aware GNN could track the reduction process and efficiently extract appropriate indepedent sets. One must note that large reduction (above 50%) may sometimes increase color gap $(OPT(G) - ALG(G))$ as the graph may

Table 4: Numerical results for Guided beam search with different metrics.

| Graph Name | $\chi$ | Random | MIS-Color | $M_{ed}$ | $M_{edc}$ | $M_{ce}$ | $M_{acc}$ | $M_w$ |
|---|---|---|---|---|---|---|---|---|
| fpsol2.i.1 | 65 | 65.6±0.5 | 65.6±0.55 | 65±0 | 65.2±0.4 | 65.2±0.4 | 65.2±0.4 | 65.4±0.5 |
| queen13_13 | 13 | 16.2±0.8 | 16±0.7 | 15±0 | 15±0 | 16.4±0.5 | 15±0 | 15±0 |
| DSJC500.9 | ? | 154.2±2.58 | 148.2±3.11 | 142.8±1.9 | 142±2.1 | 155.4±1.5 | 141±0.7 | 144.4±3.4 |
| zeroin.i.3 | 30 | 31.2±0.4 | 31±0.7 | 31±0 | 31±0 | 30.4±0.5 | 31±0 | 30.4±0.5 |
| DSJC250.9 | ? | 88.4±1.9 | 86±1.22 | 80±1 | 81±1.4 | 87.8±1.3 | 80.2±1.9 | 80.6±0.5 |
| DSJC125.5 | ? | 22.2±1.3 | 21.6±1.14 | 20.4±0.5 | 20.2±0.4 | 21.8±0.4 | 20.2±0.4 | 20.6±0.5 |
| le450_5d | 5 | 5.8±0.8 | 5.8±0.45 | 5.8±0.8 | 5.8±0.4 | 5.6±0.5 | 6.4±0.8 | 5.6±0.5 |

Table 5: Results for value aware guided beam search.

| Graph Name | $\chi$ | Nodes | Edges | Nodes (Red.) | Edges (Red.) | GBS | Time (sec) | Value-Time GBS | (sec) |
|---|---|---|---|---|---|---|---|---|---|
| homer | 13 | 556 | 1628 | 315 | 1331 | 14 | 250.34 | 14 | 100.54 |
| huck | 11 | 74 | 301 | 68 | 295 | 11 | 20.37 | 11 | 23.48 |
| david | 11 | 87 | 406 | 77 | 396 | 11 | 23.64 | 11 | 24.11 |
| anna | 11 | 138 | 493 | 112 | 467 | 12 | 42.06 | 12 | 45.12 |
| miles250 | 8 | 125 | 387 | 118 | 380 | 9 | 45.65 | 8 | 35.85 |
| jean | 10 | 77 | 254 | 55 | 229 | 10 | 35.20 | 10 | 13.47 |
| Cora | 5 | 2708 | 5278 | 1632 | 3966 | 7 | 7.34 | 7 | 3.37 |
| Citeseer | 6 | 3327 | 4552 | 1436 | 2384 | 8 | 17.65 | 8 | 3.32 |
| Pubmed(75%) | 8 | 19717 | 44324 | 5646 | 23815 | 10 | 542.46 | 15 | 20.82 |
| Pubmed(60%) | 8 | 19717 | 44324 | 7885 | 30139 | 10 | 542.46 | 14 | 79.38 |
| Pubmed(50%) | 8 | 19717 | 44324 | 9857 | 34084 | 10 | 542.46 | 13 | 120.83 |
| Pubmed(40%) | 8 | 19717 | 44324 | 11830 | 36437 | 10 | 542.46 | 13 | 130.89 |
| Pubmed(25%) | 8 | 19717 | 44324 | 14787 | 39394 | 10 | 542.46 | 13 | 258.55 |
| Ego Facebook | ? | 4039 | 88234 | 3962 | 88157 | 92 | 175.38 | 88 | 73.89 |

become much more complex like it happens in case of Pubmed (refer table 5). Table 6 highlights how the select and nonselect values of nodes behave with increase in reduction. For reduction beyond 50%, we observe a sharp increase in select and non-select value. This means that such large number of nodes are clustered together and say if this vertex in reduced graph with non-select value 1554 is not selected then in original graph 1554 vertices are inserted together in original MIS. Note that such a large MIS extracted may not be efficient for coloring always as it could lead to use of more colors for coloring remaining graph. Thus, a reduction stopping criterion can be defined for large graphs based on the select/non-select values: the reduction process is terminated once either value exceeds approximately $4\% - 5\%$ of the number of nodes in the original graph.

| Reduction % | Nodes | Edges | Nodes (Red.) | Edges (Red.) | Max select value | Max nonselect value |
|---|---|---|---|---|---|---|
| 75% | 19717 | 44324 | 5646 | 23815 | 328 | 1554 |
| 60% | 19717 | 44324 | 7885 | 30139 | 96 | 677 |
| 50% | 19717 | 44324 | 9857 | 34084 | 14 | 76 |
| 40% | 19717 | 44324 | 11830 | 36437 | 7 | 34 |
| 25% | 19717 | 44324 | 14787 | 39394 | 4 | 23 |

Table 6: Maximum select and non-select values for reduced Pubmed graph with different reduction (%). Higher values indicates that nodes are getting clustered together due to reduction which can impact coloring.

Also, for these large graphs, the presence of multiple MIS solutions could confuse the generation of MIS in starting states. While constructing MIS from probability maps (refer algorithm 4), the algorithm assigns

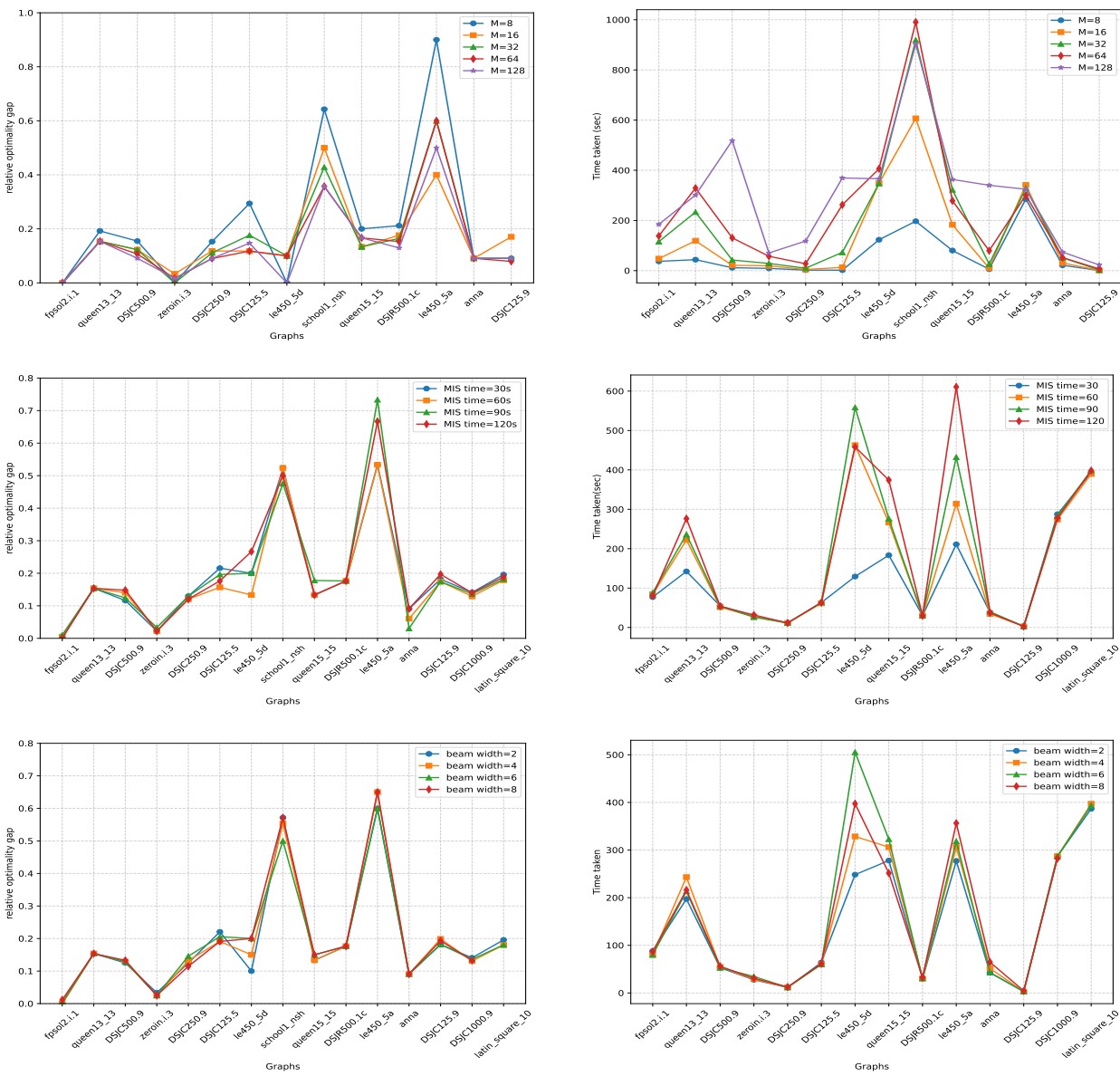

Figure 5: The figure shows how relative optimality gap and computational cost is affected by different parameters such as number of MIS probability maps used(M), MIS search time and beam width.

labels 1/0 (for chosen/not-chosen) based on probability maps and breaks the loop once it hits an already labeled vertex. This semi-labeled graph is added to queue and process of labeling is further repeated for remaining graph. But for such large graphs the algorithm might never hit a completely labeled graph state in time given for MIS generation. Thus, to ensure MIS generation, we relaxed the labeling criterion by replacing break with continue in algorithm 3 and now picking only 3/4th of vertices with good probability scores in descending order. We refer this method as C-GBS. Hence, for large graphs in table 5 the results are obtained using C-GBS.

## 7.5 Ablation Study and Impact of Parameters

GBS algorithm makes use of several parameter values such as beam width, number of MIS probability maps and MIS search time budget. Here, we study the impact of these parameters on solution quality as well as

on computational cost required. The metric used to measure solution quality is relative optimality gap that is computed as

$$relative\_opt\_gap(G) = \frac{\chi_{alg}(G) - \chi^*(G)}{\chi^*(G)}$$

where $\chi_{alg}(G)$ denotes minimum number of colors used by algorithm that is being tested and $\chi^*(G)$ is the optimal/ best-known value of chromatic number for graph $G$. Figure 5 illustrates the impact of different parameter settings on solution quality and computational time across the tested graph instances.

The beam width determines the extent of solution-space exploration performed by the GBS algorithm. However, as we can observe in figure 5, relative optimality gap remains consistent with changing beam width value. This indicates that the GBS algorithm guided by suitable metrics is able to correctly identify independent sets relevant for coloring without requiring large beam size for exploration. Hence, empirically a beam of width $c = 4$ or $c = 6$ is sufficient to obtain good quality solutions.

Next, we analyze how MIS search time could effect solution quality. Increasing the MIS search time will allow the MIS extraction procedure within GBS to identify larger independent sets. However, in the context of graph coloring obtaining a maximum independent set is not essential. It is sufficient to generate maximal independent sets for constructing feasible color classes. Thus, allocating additional time to the MIS search will primarily increases the computational cost of the algorithm while providing only limited improvements in solution quality as can be observed from Figure 5. Thus, empirically MIS search time of $60s$ is sufficient for GBS to extract MIS required for coloring.

For the parameter, number of probability maps (M) generated, the results (in Figure 5) indicate that increasing $M$ leads to improved solution quality. Higher value of $M$ helps in capturing the multi-modal nature of the problem and allows to learn informative likelihood maps for generating MIS. However, this improvement might come at the cost of increased computational effort. Therefore, choosing $M = 32$ or $64$ helps in achieving optimality as well as computational efficiency.

## 7.6 Runtime analysis

Consider graph $G(V, E)$ with $|V| = n$ and $|E| = m$ and queue capacity $c$ given as input to GBS algorithm. Let $k$ be the maximum number of MIS extracted for a graph state. For each processed state the algorithm computes maximum $k$ MIS (in time $T_{MIS}(n, m)$), build $k$ candidate graphs (in time $k.O(n+m)$) and update queue (in time $O(k.log(c))$). The number of processed states can be at most $c.n$. The $find\_multiple\_MIS(.)$ function requires one GNN forward pass ($O(n + m)$) to generate all probability maps and further exploration of partial solutions which is controlled by giving time budget. In practice, the running time scales approximately linearly with graph size for the majority of tested graphs. Runtime comparison cannot be drawn with GNN-based solvers as all existent solvers use optimal/near-optimal colors for coloring and find number of conflicts. Obtaining feasible coloring from such a coloring obtained is not specified which can take additional time. Moreover, in worst case we may have to introduce as many colors as number of conflicts which will significantly increase the number of colors used. However, we have drawn comparison between runtime of GBS and W-GBS for groups of graphs with different edge densities (refer figure 6). We have also reported runtime for Value-GBS in table 5.

## 7.7 Circuit Depth Optimization with GBS

We demonstrate how GBS can be used to optimize circuit depth by testing the algorithm on benchmark quantum circuits (Li et al., 2022a; Chen et al.). Direct comparison with qiskit built-in function to optimize circuits is limited because Qiskit transpiler does not work as a gate ordering solver. Thus, we test against Qiskit DAG method. This method uses qiskit library to convert a given quantum circuit to a directed acyclic graph (DAG) and then find circuit depth as length of longest directed path. This method simply computes depth for the given circuit ordering. Our method on the other hand exploits commutativity relations and builds a mixed graph that is commutativity-aware and thus could find a suitable gate ordering that can reduce depth. The tested circuit instances include both reversible arithmetic circuits such as adders, multipliers, and square-root implementations and randomly generated Clifford circuits. Arithmetic

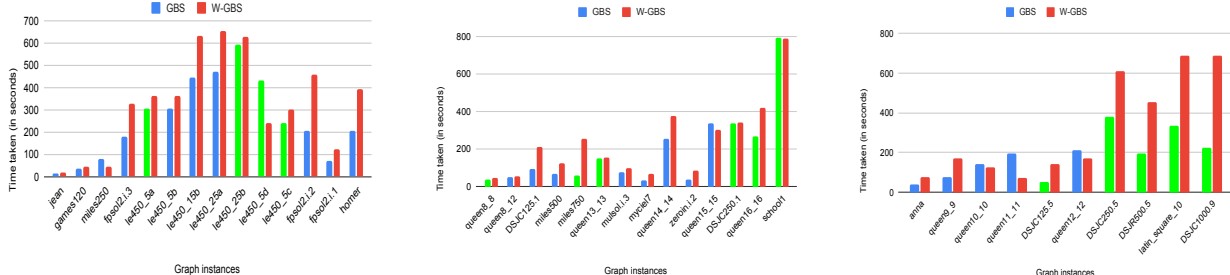

Figure 6: For better analysis the graph dataset is divided into three groups based on edge density, $G1$ with edge density $p < 0.1$ (left figure), $G2$ with edge density $0.1 \leq p < 0.5$ (middle figure), $G3$ with edge density $p \geq 0.5$ (right figure). In each figure, graphs are arranged in increasing order of number of vertices. Graphs for which W-GBS performed better than GBS are highlighted with green color.

| Circuit | Nodes | Edges | Qiskit DAG | GBS |
|---|---|---|---|---|
| GROVER_N2. | 18 | 93 | 12 | 12 |
| MULTIPLIER_N15 | 73 | 947 | 49 | 49 |
| MULTIPLIER_N15_TRANS | 563 | 29767 | 257 | **213** |
| ADDER_N64 | 268 | 1696 | 78 | **53** |
| ADDER_N64_TRANS | 1220 | 26002 | 462 | **391** |
| ADDER_N28 | 116 | 724 | 42 | **33** |
| SQUARE_ROOT_N18 | 558 | 23131 | 203 | **190** |

Figure 7: In the given table, we compare circuit depth between qiskit DAG method and GBS on QASM benchmark instances. The graph shows the results obtained for random Clifford circuits.

circuits are particularly challenging for scheduling because they typically contain dense networks of CNOT and Toffoli gates. Although these networks give rise to complex dependency structures, these circuits often contain commuting gate subsets that can be exploited to increase parallelism. We also test on random Clifford circuits which are widely used in quantum benchmarking and compilation studies.

Results obtained on these instances are reported in Figure 7. Achieving depth reduction on these circuit instances demonstrate that the proposed GBS-based scheduling approach is capable of uncovering hidden parallelism in diverse quantum circuit classes. Another advantage of the proposed approach is that it reduces circuit depth solely through gate reordering, without modifying the circuit functionality or introducing additional gates. However, one must note that gate reordering alone cannot reduce the depth of every quantum circuit. For some instances, such as grover_n2 (in Figure 7), the circuit structure may not allow any parallelism further. Nevertheless, these results indicate that many practical circuits contain significant latent parallelism that can be effectively exploited by this commutativity-aware gate reordering approach.

## 8 Conclusion

In this work, we address the classical graph coloring problem (GCP) through a learning-based approach that integrates GNNs with guided beam search (GBS). A key idea throughout this work is the repeated extraction of MIS to build up color classes. The structured exploration capability of beam search guided by well-curated metrics enables the algorithm to generate high-quality color assignments for complex graph instances. We also introduce value-aware GNN that helps to integrate reduction techniques with GBS algorithm and enables scalable inference on larger graphs. This idea of constructing coloring by sequentially

extracting independent sets proves to be beneficial for lot of applications like the quantum circuit depth optimization that we thoroughly discussed. While the proposed method demonstrates promising results, several directions remain open for further exploration. For instance, alternative metrics can be explored that could offer better heuristics or problem-specific insights. In addition, domain-specific metrics could be incorporated when applying graph coloring to practical problems such as scheduling, register allocation or quantum circuit mapping where additional constraints or priorities exist beyond minimizing the number of colors. For scaling purposes, one could design reduction strategies that can operate effectively on dense graphs especially those with high clustering or global connectivity. Overall, the proposed framework highlights the potential of learning-based approaches for graph coloring and illustrates how learning-based solvers can be adapted to operate effectively on reduced graph representations. We hope that this work encourages further exploration of learning-based approaches for scalable graph coloring and related combinatorial optimization problems.

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

---

**Algorithm 3** Basic MIS

---

**Input:** Graph $\mathcal{G}$
**Output:** Labelling of all the vertices of $\mathcal{G}$
$v \leftarrow \text{argsort}(f(\mathcal{G}; \theta))$ in a descending order
**for** $i := 1$ to $||v||$ **do**
    **if** vertex $v(i)$ is already labelled in $\mathcal{G}$ **then**
      break
    **end if**
    Label $v(i)$ as 1 in $\mathcal{G}$
    Label the neighbors of $v(i)$ as 0 in $\mathcal{G}$
**end for**
$\mathcal{G}' \leftarrow$ residual graph of $(G)$ by removing labelled vertices
**if** $\mathcal{G}'$ is not empty **then**
    Run BasicMIS on $\mathcal{G}'$
**end if**

---

# A   Appendix

## A.1   Finding MIS using probability map

The approach used to learn $find\_multiple\_MIS(.)$ function is from the paper given by Li et al. (2018b) where they find maximum independent set (MIS) for a graph $G = (V, E)$ by estimating the likelihood for each vertex $v \in V$ of being present in MIS. The network learned uses GCN architecture and takes the graph $G$ on $N$ vertices as input and generates multiple probability maps $\langle f^1, f^2, \ldots, f^M \rangle$ where each probability map on vertices $p : V \to [0, 1]^N$ indicates how likely each vertex is to belong to the MIS. If only a single map is generated then Algorithm 3 is used. The vertices are sorted based on their likelihood to be in MIS and labeled in this order. When a vertex is labelled 1 all of its neighbors are labelled 0. However, if the algorithm hits an already labelled vertex then this labelling is not changed. Instead, the likelihood map is generated again for the remaining unlabeled graph and the process is repeated until all vertices get labelled. Now, in case of multiple probability maps, multiple MIS are generated using Algorithm 4 which uses breadth-first approach. For each probability map $f^i(G)$ the vertices are sorted in descending order of likelihood. The vertices are then labeled 1 with its neighbors 0. This process stops when the next vertex in the sorted list is already labeled as 0. Then a residual graph $G'$ is obtained by removing all the labeled vertices and added in queue for further exploration. The exploration of all the intermediate states is further controlled by a time budget. For executing **C-GBS** as discussed in results section, break in line 10 of algorithm 4 is replaced by continue. Also, in line 7 instead of considering all vertices only top $t$ vertices (sorted in descending order of probability map) are considered with $t$ in range $25\% - 75\%$.

## A.2   Experimental setup

**Datasets:** For training the MIS extracting GCN we use 1000 random graphs generated using the Erdos–Renyi(ER) model $G(n, p)$ where $n$ (number of nodes) is randomly chosen between 10 and 300 and $p$ (probability of edge formation) is randomly chosen between 0.01 and $min\left(0.1, \frac{5}{n}\right)$. $p$ is chosen to ensure that the graphs are sparse for large $n$ keeping it manageable.

**Training:** We generate multiple labeling solutions for the graphs in training dataset using integer linear programming(ILP) solver called Gurobi(Gurobi Optimization, Inc., 2025). Corresponding to each graph we have a list of binary label vectors, each vector representing one possible MIS solution for that graph. All of these label vectors have same size. We use Adam optimizer with single-graph mini-batches and learning rate $10^{-4}$. The loss function used is hindsight loss which captures loss from all the $M$ probability maps generated.

**Network settings:** The network settings for training MIS extraction GCN remains same as in paper by Li et al. (2018b). The network has $L = 20$ GCN layers and the input feature vector for every vertex is set to **1**

---

**Algorithm 4** MIS with diversity and tree search

---

**Input:** Graph $\mathcal{G}$
**Output:** The best solution found
Initialize queue $\mathcal{Q}$ with $\mathcal{G}$
**while** execution time is under budget **do**
    $\mathcal{G}' \leftarrow \mathcal{Q}.\text{random\_pop}()$
    **for** $m := 1$ to $M$ **do**
        $\mathbf{v} \leftarrow \text{argsort}(f^m(\mathcal{G}_t; \theta))$ in descending order
        **for** $i := 1$ to $\|\mathbf{v}\|$ **do**
            **if** vertex $\mathbf{v}(i)$ is already labeled in $\mathcal{G}'$ **then**
                **break**
            **end if**
            Label $\mathbf{v}(i)$ as 1 in $\mathcal{G}'$
            Label the neighbors of $\mathbf{v}(i)$ as 0 in $\mathcal{G}'$
        **end for**
        **if** $\mathcal{G}'$ is completely labeled **then**
            Update the current list of best solutions
        **else**
            Remove labeled vertices from $\mathcal{G}'$
            Add $\mathcal{G}'$ to $\mathcal{Q}$
        **end if**
    **end for**
**end while**

---

(vector having all entries equal to 1) and is of size $C_0 = 32$. The widths of the hidden layers are set identical to $C_l = 32$. The width of the output layer $C_L = M$, where $M$ is the number of output probability maps. We use $M = 32$. For the Beam Search algorithm, we experimented with different values of the queue capacity $c$ and found that $c = 4$ provides a good balance between computational cost and performance. Increasing $c$ beyond this value results in negligible improvements. Also, time budget set for predicting MIS is $T = 60s$. All the methods are trained with 2×Nvidia A100 card GPUs and evaluated on a single NVIDIA A100 GPU with 2× Intel Xeon Platinum 8358 (32 cores 2.6 GHz) CPU.

**Baselines:** We compare the presented approach to several traditional heuristics based algorithms and ML-solvers developed for graph coloring. The heuristics based algorithms used for comparison are LF, SL, DSatur (Brélaz, 1979),RLF (Leighton, 1979),Tabucol (Hertz & de Werra, 1987)

- **LF:** Vertices are ordered in decreasing order of degree and then colored sequentially using the smallest available color not used by their colored neighbors.

- **SL:** The vertices with the smallest degree are removed repeatedly and then coloring is done in reverse order of removal.

- **DSatur:** The algorithm (Brélaz, 1979) begins by coloring the vertex with the highest degree and then at each step an uncolored vertex with the highest saturation degree is selected for coloring. Here, saturation degree of a vertex is the number of different colors used by its already-colored neighbors.

- **RLF:** The algorithm (Leighton, 1979) tries to form large independent sets iteratively producing fewer colors. To form independent set, a vertex of maximum degree is chosen followed by iteratively adding vertices that are not adjacent to any vertex in the current color class but are most adjacent to vertices outside it.

- **Tabucol:** is a tabu search–based algorithm (Hertz & de Werra, 1987) designed to minimize the number of color conflicts in a graph.It starts from an initial (possibly infeasible) coloring and itera-

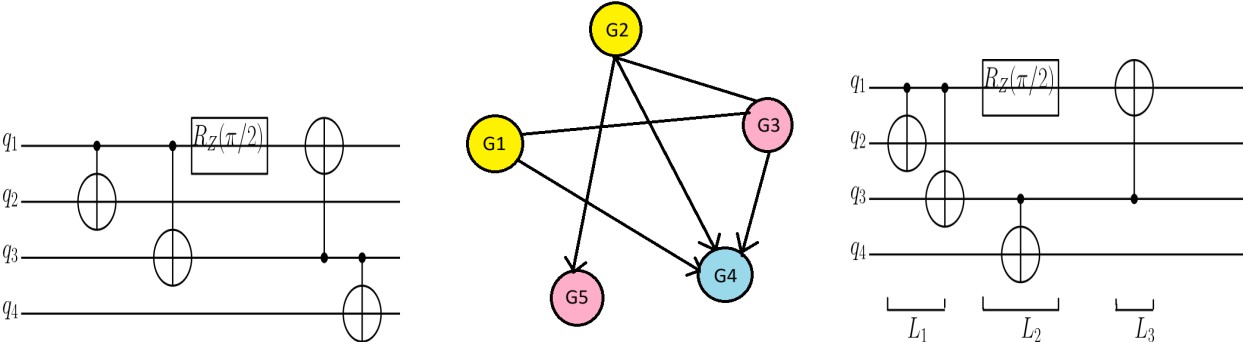

Figure 8: Sample circuit $C$ ($depth = 5$), its corresponding dependency graph $G$ and the optimized circuit $C'$ ($depth = 3$).

tively improves it by recoloring vertices while avoiding cycles and local minima through the use of a tabu list.

### A.3 Dependency graph construction for Quantum Circuit

In this section we explain in detail how a dependency graph is generated from a quantum circuit and further how the circuit is optimized using optimal coloring. Consider the sample circuit $C$ given in figure8, it consist of 4 qubits and 5 gates. To generate the corresponding dependency graph, the gates are labeled as follows:-

- $G1$: $CNOT(q_1, q_2)$

- $G2$: $CNOT(q_1, q_3)$

- $G3$: $R_Z(\pi/2)(q_1)$

- $G4$: $CNOT(q_3, q_1)$

- $G5$: $CNOT(q_3, q_4)$

Now, to obtain coloring for this mixed graph, we first obtain a corresponding undirected graph $G_u$ selecting nodes having indegree equal to 0. Thus, $G_u$ consist of nodes $G1, G2$ and $G3$. The algorithm chooses MIS $\{G1, G2\}$. Next this MIS is removed and now indegree of $G5$ becomes 0. Thus, in next iteration the subgraph with nodes $G3$ and $G5$ is selected and algorithm obtains MIS $\{G3, G5\}$. Finally node $G4$ is left which is colored with a new color. Thus, coloring obtained uses three colors as shown in figure 8. Nodes with same color labels can be executed in parallel and we obtain an optimized circuit with depth 3.

For comparison we obtain circuit depth for test instances using Qiskit. Qiskit is an open-source quantum computing software development kit used to design, simulate, compile, and run quantum programs on both simulators and real quantum hardware. In Qiskit, the DAG depth method is a way to compute the circuit depth by first converting a quantum circuit into a Directed Acyclic Graph (DAG) and then finding the length of the longest dependency path in that graph. For the sample circuit $C$ we obtain DAG given in figure 9.

The circuit consist of two types of gates $CNOT(.,.)$ and single-qubit rotation gate $R_Z(\pi/2)(.)$. For a $CNOT(q_1, q_2)$ gate, $q_1$ is control qubit and $q_2$ is target qubit. $CNOT$ gates can commute only if they have disjoint qubits or same control and different targets. Thus, we obtain the corresponding dependency graph $G$ with five vertices and following edges:-

- $e_1$: undirected edge between $G1$ and $G3$ because the two gates commute and share qubit $q1$.

- $e_2$: directed edge $(G1, G4)$ because the two gates cannot commute as $q1$ is control qubit for $G1$ and target qubit for $G4$.

- $e_3$: undirected edge between $G2$ and $G3$ because the two gates commute and share qubit $q1$.

- $e_4$: directed edge $(G2, G4)$ because the two gates cannot commute as $q1$ is control qubit for $G2$ and target qubit for $G4$.

- $e_5$: directed edge $(G2, G5)$ because the two gates cannot commute as $q3$ is control qubit for $G5$ and target qubit for $G2$.

- $e_6$: directed edge $(G3, G4)$ as the two gates do not commute.

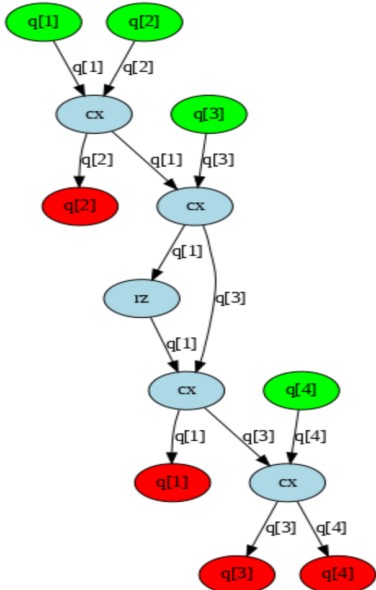

Figure 9: Directed acyclic graph (DAG) obtained for Sample circuit C in figure 8 using qiskit.

### A.4 Quantum circuit depth optimization algorithm

**Theorem A.1.** *The optimized circuit $C'$ obtained from above algorithm is executable that is the coloring obtained is suitable.*

**Proof:** Let $\phi$ be the coloring obtained from the algorithm. For each directed edge $(u, v) \in E_d$, $v$ will have indegree 0 only if $u$ is chosen in some independent set that is $u$ is colored first. Thus, $\phi(u) < \phi(v)$ holds for coloring $\phi$. Also for each undirected edge $\{u, v\} \in E_u$ either $u$ and $v$ belongs to $G_{exec}$ or not. If not then they will always receive distinct colors. If both are present simultaneously in $G_{exec}$ then also they will not be selected together in an independent set. Thus, $\phi(u) \neq \phi(v)$ holds and we can conclude that coloring $\phi$ is suitable and thus the circuit $C'$ obtained from $\phi$ will be executable.

---

**Algorithm 5** Guided Beam Search for Quantum circuit depth optimization

---

**Input:** Quantum circuit $C$, queue capacity $c$
**Output:** Quantum Circuit $C'$ with minimized circuit depth
Generate dependency graph $G(V, E_d, E_u)$ from circuit $C$
Initialize queue $Q$ with capacity $c$
Initialize $min\_colors \leftarrow \infty$
Enqueue $(G, 0)$ into $Q$
**while** $Q$ is not empty **do**
    $(G_{current}, num\_colors) \leftarrow$ Dequeue from $Q$
    Find $G_{exec}(V', E_u')$ from $G_{current}$ where $V'$ are vertices with indegree 0
    and $E_u'$ are undirected edges between them
    **if** $G_{exec}$ has no vertices **then**
        $min\_colors \leftarrow \min(min\_colors, num\_colors)$
        continue
    **end if**
    $MIS\_list \leftarrow$ find_multiple_MIS$(G_{exec})$
    **for** each $MIS$ in $MIS\_list$ **do**
        $G_{new} \leftarrow G_{current}$ with $MIS$ removed
        Enqueue $(G_{new}, num\_colors + 1)$ into $Q$
    **end for**
    update_queue$(Q)$
**end while**
Construct circuit $C'$ by arranging gates in sequence of increasing color label.
Gates with same color label are arranged in parallel.
**return** Circuit $C'$

---

## A.5 Results for DIMACS dataset

Table 7: Coloring results and comparison with traditional heuristic algorithms. Number of colors used by optimal coloring obtained from each solver is reported. Metric used for GBS is edge density with colors used.

| Graph Name | $|V|$ | $|E|$ | $\chi$ | GBS | W-GBS | RLF | DSatur | LF | SL |
|---|---|---|---|---|---|---|---|---|---|
| jean | 80 | 254 | 10 | 10 | 10 | 10 | 10 | 10 | 10 |
| games120 | 120 | 638 | 9 | 9 | 9 | 9 | 9 | 9 | 9 |
| miles250 | 128 | 387 | 8 | 9 | 8 | 8 | 8 | 8 | 8 |
| fpsol2.i.3.col | 425 | 8688 | 30 | 30 | 31 | 30 | 30 | 30 | 30 |
| le450_5a | 450 | 5714 | 5 | 8 | 7 | 8 | 10 | 11 | 11 |
| le450_5b | 450 | 5734 | 5 | 8 | 8 | 8 | 9 | 12 | 13 |
| le450_25a | 450 | 8260 | 25 | 27 | 27 | 25 | 25 | 25 | 25 |
| le450_25b | 450 | 8263 | 25 | 27 | 27 | 25 | 25 | 25 | 25 |
| le450_5d | 450 | 9757 | 5 | 5 | 5 | 5 | 12 | 14 | 11 |
| le450_5c | 450 | 9803 | 5 | 5 | 5 | 5 | 10 | 12 | 12 |
| fpsol2.i.2.col | 451 | 8691 | 30 | 30 | 31 | 30 | 30 | 30 | 30 |
| fpsol2.i.1.col | 496 | 11654 | 65 | 65 | 65 | 65 | 65 | 65 | 65 |
| homer | 561 | 1629 | 13 | 13 | 13 | 13 | 13 | 13 | 13 |
| queen7_7 | 49 | 952 | 7 | 7 | 7 | 9 | 11 | 13 | 12 |
| queen8_8 | 64 | 728 | 9 | 10 | 9 | 11 | 12 | 13 | 14 |
| queen8_12 | 96 | 1368 | 12 | 13 | 13 | 13 | 14 | 16 | 16 |
| DSJC125.1 | 125 | 1472 | ? | 6 | 5 | 6 | 6 | 7 | 7 |
| miles500 | 128 | 1170 | 20 | 21 | 20 | 20 | 20 | 20 | 20 |
| miles750 | 128 | 2113 | 31 | 32 | 32 | 31 | 31 | 32 | 31 |
| queen13_13 | 169 | 6656 | 13 | 15 | 15 | 16 | 17 | 22 | 20 |
| mulsol.i.3.col | 184 | 3916 | 31 | 31 | 31 | 31 | 31 | 31 | 31 |
| myciel7 | 191 | 2360 | 8 | 8 | 8 | 8 | 8 | 8 | 8 |
| queen14_14 | 196 | 8372 | ? | 16 | 16 | 17 | 19 | 23 | 22 |
| zeroin.i.2.col | 211 | 3541 | 30 | 30 | 30 | 30 | 30 | 30 | 30 |
| queen15_15 | 225 | 10360 | ? | 17 | 17 | 19 | 21 | 25 | 23 |
| DSJC250.1 | 250 | 6436 | ? | 10 | 10 | 10 | 10 | 11 | 12 |
| queen16_16 | 256 | 12640 | ? | 19 | 18 | 19 | 23 | 26 | 25 |
| school1 | 385 | 19095 | ? | 19 | 18 | 27 | 17 | 32 | 15 |
| anna | 38 | 493 | 11 | 12 | 11 | 11 | 11 | 11 | 11 |
| queen9_9 | 81 | 2112 | 10 | 10 | 10 | 12 | 13 | 15 | 15 |
| queen10_10 | 100 | 2940 | ? | 11 | 11 | 13 | 14 | 16 | 16 |
| queen11_11 | 11 | 3960 | 11 | 13 | 13 | 14 | 15 | 19 | 18 |
| DSJC125.5 | 125 | 7782 | ? | 20 | 20 | 20 | 22 | 23 | 25 |
| queen12_12 | 144 | 5192 | ? | 14 | 14 | 15 | 16 | 20 | 19 |
| queen16_16 | 256 | 12640 | ? | 19 | 18 | 19 | 23 | 26 | 25 |
| DSJC250.5 | 250 | 31366 | ? | 33 | 31 | 34 | 37 | 41 | 39 |
| DSJC500.5 | 500 | 62624 | ? | 54 | **53** | 60 | 65 | 71 | 70 |
| DSJC1000.1 | 1000 | 49629 | ? | 25 | 24 | 24 | 27 | 29 | 30 |
| DSJC1000.5 | 1000 | 249826 | ? | 95 | 94 | 107 | 115 | 121 | 125 |

