# OpenReview forum: "Graph Coloring via Learning and Metric-Guided Independent Set Extraction"
_TMLR — Under review for TMLR_

### Review · Reviewer_SYHt · 2026-05-06

**Summary Of Contributions:**

The paper aims to provide a graph coloring algorithm by repeatedly extracting large independent sets using GNN-guided search, metric-based beam selection, and graph reductions to improve scalability. Specifically, the paper proposes Guided Beam Search (GBS), which colors a graph by iteratively selecting independent sets as color classes. A GNN-based maximum independent set extractor generates multiple candidate independent sets, and a beam-search queue retains only the most promising partial colorings based on structural metrics such as edge density, edge density adjusted by the colors used, closeness centrality, color efficiency, and progress rate. The paper further introduces Weighted-GBS, where a lightweight neural network predicts graph-specific weights for combining metrics based on graph features such as size, density, degree statistics, clustering coefficient, and diameter. To improve scalability, the paper introduces a value-aware GNN for reduced graphs via vertex and pendant folding. Each reduced node stores select and nonselect values so the model can estimate how choices in the reduced graph translate to independent-set quality in the original graph. The method is evaluated on DIMACS coloring benchmarks, citation networks, social graphs, and quantum circuit depth-optimization tasks.

**Audience:**

Yes

**Audience Explanation:**

Graph coloring is absolutely fundamental for Graph Neural Networks, and algorithms to estimate chromatic numbers or other graph attributes can be very useful for GNNs. However, some of these insights could be very niche or of interest to ML theory subcommunities.

**Broader Impact Concerns:**

I don't see a direct concern. However, since the application presented includes social networks (including citation nets), a small comment on potential uses or applications could enhance the broad adoption of the algorithm presented in this paper.

**Claims And Evidence:**

No

**Claims Explanation:**

The paper is quite interesting. It combines learning with a classical combinatorial structure: independent-set extraction as color-class construction. The empirical scope is reasonably broad, covering DIMACS graphs, citation networks, social graphs, and quantum circuits; and the value-aware reduction idea seems effective.

The paper is mostly empirically sound as a heuristic framework. The central idea, that repeated independent-set extraction produces color classes, is interesting. The quantum-circuit suitability proof is plausible because directed dependencies are handled by only coloring zero-indegree executable gates, and undirected conflicts are handled through independent sets. However, the paper does not provide approximation guarantees, and the experimental design has some weaknesses: baseline comparisons mix different output formats, hyperparameter sensitivity is limited, and some reported results show that reductions can substantially degrade solution quality.

In Table 2, the proposed GBS method reports the number of colors used, whereas most competing GNN methods (PI-GCN, GDN, GNN-NU, PI-SAGE, GIN, and Tabucol) report the number of conflicts while using a fixed number of colors equal to the known or presumed chromatic number $\chi(G)$. For example:

* On queen13_13, GBS uses 15 colors, while GNN-NU reports 15 conflicts using only 13 colors.
* On queen11_11, GBS uses 13 colors, while GNN-NU reports 13 conflicts at 11 colors.

These are not directly comparable quantities. A coloring with slightly more colors but zero conflicts is qualitatively different from an infeasible coloring with fewer colors but unresolved conflicts. The paper acknowledges this partially in the runtime discussion, noting that other methods “use optimal/near-optimal colors and find the number of conflicts,” but it does not normalize the comparison through a consistent feasible-coloring conversion procedure.

This creates some concerns. First, it is unclear how many additional colors the conflict-based methods would ultimately need after repair/postprocessing. Second, the comparison may either favor GBS unfairly (because it outputs feasible colorings directly) or disadvantage it (because other methods are solving a harder constrained problem). Third, a reader cannot determine whether GBS truly dominates prior methods on the same optimization target.

Another weakness is that the paper introduces many design choices and hyperparameters, but provides only minimal analysis of their sensitivity. Examples include: beam width c, number of MIS probability maps M, MIS search time budget, metric selection, weighted metric coefficients, graph reduction thresholds, and the modified “C-GBS” relaxation heuristic. However, no ablation plots or sensitivity curves are shown. Also, any metric analysis should not only report final coloring outcomes, but also robustness or variance across runs.


Finally, the value-aware reduction framework is one of the paper’s more interesting contributions, but the experimental results reveal a substantial tradeoff between scalability and coloring quality. The paper itself explicitly acknowledges “a large reduction (above 50%) may increase the color gap.” This issue appears clearly in Table 3. For the PubMed graph, the reductions dramatically worsen solution quality despite large runtime gains.

Note that the paper explains that aggressive reductions may make the reduced graph “structurally complex and dense,” distort MIS structure, and increase ambiguity in the probability maps. Furthermore, as shown in Figure 2, the reduced graph becomes harder to color than the original. However, the paper positions reductions as a scalability mechanism, yet scalability improvements come at potentially severe degradation in solution quality, and it does not characterize when reductions are safe versus harmful.

**Requested Changes:**

Please consider providing a stronger evaluation. For instance, you could convert all outputs into feasible colorings, report both conflicts and final colors, and compare under fixed time/repair budgets.

For the second weakness I outlined above, I would suggest incorporating ablation studies and confidence intervals across runs.

Finally, to address the third weakness listed above (the lack of characterization of the trade-off between reduction and color representation), I would suggest including at least one of the following: the theoretical conditions under which reductions preserve color structure, an adaptive reduction-stopping criterion, or a learned predictor that estimates whether reduction will degrade performance.

---

> ### Author Response · Authors · 2026-06-16
> **Response to Reviewer SYHt**
>
> We sincerely thank the reviewer for the careful reading of the manuscript and for the constructive suggestions. Below we address each comment in turn and summarize the corresponding revisions.
>
> (1) **Comment:**  "In Table 2, the proposed GBS method reports the number of colors used, whereas most competing GNN methods report the number of conflicts while using a fixed number of colors equal to the known or presumed chromatic number. These are not directly comparable quantities...."
>
> **Revision:** To support proper comparison we know report (in table 3) the minimum number of colors required to obtain a proper coloring (coloring with zero conflicts) for ML-solvers such as PI-GCN, PI-SAGE, GNN-NU and GIN. This is done by running these methods for different values of k. The minimum k that generates a conflict-free coloring is reported. Due to the unavailability of the source code, we have to remove GDN from comparison table. Also, we would like to compare only ML-based solvers in this table so we have removed the Tabucol algorithm. Tabucol algorithm is a metaheuristic based algorithm that dates back to 1987 and now a lot of improved heuristics are available for comparison which we report in table 2.
>
> (2) **Comment:**  "Another weakness is that the paper introduces many design choices and hyperparameters, but provides only minimal analysis of their sensitivity. Examples include: beam width c, number of MIS probability maps M, MIS search time budget, metric selection, weighted metric coefficients, graph reduction thresholds, and the modified “C-GBS” relaxation heuristic. However, no ablation plots or sensitivity curves are shown. Also, any metric analysis should not only report final coloring outcomes, but also robustness or variance across runs."
>
> **Revision:** We understand that for proper analysis of framework it is important to properly address the impact of hyper parameters such as beam width, MIS search time, number of probability maps used etc. Thus, we have updated experiment section and have now included ablation study for all these parameters. A separate section is made to address metric analysis. Now, in table 4 we report average colors used and standard deviation obtained across multiple runs for all the metrics introduced. Also, we compare these metric results against baseline MIS+ Beam search without metric. In this method at each step, out of the multiple MIS generated (all of same size), we randomly choose c MIS where c is beam width. This is referred to as Random in the table. Also, we introduce another baseline comparison in which we repeatedly apply Li et al. MIS extraction algorithm  and color the obtained independent set. This is referred to as MIS-Color.
> All these results for metric analysis are reported in table 4. As far as metric selection is concerned, a single metric cannot perform well for all graph instances as they all are based on different structural properties of graph. However, as observed empirically weighted metric performs significantly better for most of the instances and helped in improving results. Next comes the edge density with colors used metric that also performed well on most of the instances.
>
> Also, the weighted metric coefficients are determined by a learned weight predictor that takes as input a feature vector corresponding to input graph and outputs optimal weight vector for graph G.
>
> Now, for large graphs, the presence of multiple MIS solutions could confuse the generation of MIS in starting states. While constructing MIS from probability maps (refer algorithm 3) for such large graphs the algorithm might never hit a completely labeled graph state in time given for MIS generation. Thus, to ensure MIS generation, we relaxed the labeling criterion by replacing break with continue in algorithm 3 and now picking only $3/4$th of vertices with good probability scores in descending
> order. This method is referred to as C-GBS. Hence, for large graphs in table 5 the results are obtained using C-GBS.

---

> ### Author Response · Authors · 2026-06-16
> **Response to Reviewer SYHt : Part 2**
>
> (3) **Comment:** Trade-off between reduction and color representation, "but the experimental results reveal a substantial tradeoff between scalability and coloring quality. This issue appears clearly in Table 3. For the PubMed graph, the reductions dramatically worsen solution quality despite large runtime gains."
>
> **Revision:** To analyse this tradeoff between reduction and solution quality, we consider Pubmed graph instance. Table 6 in the updated version highlights how the select and nonselect values of nodes behave with increase in reduction. For reduction beyond 50%, we observe a sharp increase in select and non-select value for vertices in reduced graph. This means that such large number of nodes are clustered together and say if this vertex in reduced graph with large select value $s$ is selected then in original graph all $s$ vertices are inserted together in original MIS. Note that such a large MIS extracted may not be efficient for coloring always as it could lead to use of more colors for coloring remaining graph. Thus, a reduction stopping criterion can be defined for large graphs based on the select/non-select values: the reduction process is terminated once either value exceeds approximately 4%-5% of the number of nodes in the original graph.
>
> Thus, in the revised version we have tried to address all the three weaknesses pointed out. For first weakness, we improved the results table and now report minimum number of colors used for all solvers for a fair comparison. For the second weakness as suggested we have tried to give ablation studies for all the parameters. Finally, to address the third weakness listed above (the lack of characterization of the trade-off between reduction and color representation) we have discussed an adaptive reduction-stopping criterion which can be used while applying reduction techniques to maintain the quality of the solutions.

---

### Review · Reviewer_F9zs · 2026-05-18

**Summary Of Contributions:**

In this paper, a GNN-based solution for optimizing the graph coloring problem is proposed. Specifically, the authors adapt the model proposed by (Li et al.,2018) for the Maximum Independent Set (MIS) problem to use it for detecting the independent sets, and this way, colour them in a second stage. For the second part of the scheme, the authors propose the Guided Beam Search algorithm, which introduces an improvement over the classic beam search. In addition, to optimize large graph problems, the authors incorporate previously published graph reduction techniques and propose a novel loss function (value-aware binary cross-entropy) that is aligned with the introduced.

Some strong points (briefly, in the boxes below expanded):

- Graph Coloring Problem (GCP) is relevant, challenging, and has many applications.
- An important branch of the ML community is aiming to optimize combinatorial problems, including GCP with Neural architectures. It is an interesting line of work.
- ML-based algorithms for Combinatorial optimization problems look to be behind metaheuristic and other paradigms in terms of either performance or efficiency, so pushing in this direction is also a good idea.


Weak points (briefly, in the boxes below expanded):

- Many fundamental references are missing.
- Unsupported claims are included.
- Optimization paper focused on achieving SOTA results.
- Weak experimental study with regard to the included algorithms.
- Weak experimentation regarding the Quantum Circuit Depth Optimization.

**Audience:**

Yes

**Audience Explanation:**

- The graph coloring problem is a very well-known NP-hard combinatorial problem, and thus, any effort to design better-performing (or more efficient) algorithms is welcome. Since 2015, a great number of papers have been published by the Machine Learning community for optimizing combinatorial problems. In this context, paradigms such as Neural Combinatorial Optimization (NCO) have been proposed, and the community is intensely working on it.

- The proposed method relies on some previous works investigated in the field of Machine Learning in recent years. Most of the works consider/involve the use of GNNs for codifying the instances and translating the problem information to a latent space code to make decisions that enable the optimization. It looks like the right direction.

- ML-based algorithms for Combinatorial optimization problems look to be behind metaheuristic and other paradigms in terms of either performance or efficiency, so pushing in this direction, trying to develop better ML algorithms, is also a good idea.

In short, the paper contains elements that are highly relevant to the machine learning community and, consequently, to the TMLR.

**Broader Impact Concerns:**

No concerns.

**Claims And Evidence:**

No

**Claims Explanation:**

- Many of the claims made in the introduction need to be supported by references or demonstrated within the paper. To illustrate what I mean, the authors state in the introduction: ‘... for problems like graph colouring, the embeddings learned by GNNs for adjacent nodes should be dissimilar and significantly apart in the embedding space...’; however, they do not provide a reference that actually supports this claim. Whilst it is true that our intuition points in that direction, it does not necessarily have to be true.

- The paper claims that "The results demonstrate the superior performance of our method compared to existing state-of-the-art approaches", but it is not supported. The proposed algorithm indeed outperforms the performance of certain algorithms, including heuristics and metaheuristics. However, such algorithms date back to 1987, and they are not the actual state-of-the-art of the Graph Coloring Problem. A quick review of the current literature reveals that there are swarm intelligence-type metaheuristic algorithms from 2023 onwards that achieve performance metrics (the minimum value of k) for DIMACS instances that are better than those reported in Table 1.

**Requested Changes:**

To recommend the paper for acceptance, it is crucial to...
1. ... improve the presentation of the framework. The paper has a problem with its focus. It appears to be focused on improving state-of-the-art results, as if that were the end goal, whilst neglecting to explain many fundamental elements. For example, the work by Li et al. (2018) forms the basis of this study, and a series of new improvements have been incorporated to enable the use of that model on the GCP. But the explanation is very poor. Similarly, the presentation of Guided Beam Search could be improved, as could the integration of the new loss function with the preceding work. The authors should present the big picture and attempt to explain the performance achieved and the design of the algorithm that enabled it.

2. ... include truly state-of-the-art algorithms, and show that the proposed framework can outperform them. This implies considering metaheuristic algorithms from 2023 onwards (also for the heuristic and metaheuristics side). The real state-of-the-art results are much better than those in Table 1.

3. ... expand the experimentation with Quantum Circuit Depth Optimization (QCDO), include additional details,  and demonstrate the relevance of the results obtained for the literature on QCDO.

---

> ### Author Response · Authors · 2026-06-16
> **Response to Reviewer F9zs : Part 1**
>
> We sincerely thank the reviewer for the careful reading of the manuscript and for the constructive suggestions. Below we address each comment in turn and summarize the corresponding revisions.
>
> (1) **Comment:**  "... improve the presentation of the framework. The paper has a problem with its focus. It appears to be focused on improving state-of-the-art results, as if that were the end goal, whilst neglecting to explain many fundamental elements...."
>
> **Revision:**  We agree that the presentation of the algorithm in the paper could have been better. For example as pointed out "the authors state in the introduction: ‘... for problems like graph coloring, the embeddings learned by GNNs for adjacent nodes should be dissimilar and significantly apart in the embedding space...’; however, they do not provide a reference that actually supports this claim." This claim was made in the paper  "Graph neural network with negative message passing and uniformity maximization for graph coloring." where the authors use this reasoning to motivate the construction of their method GNN-NU. We have cited the reference in the introduction section as well.
>
> Also, as rightly pointed based on presentation of GBS algorithm in the paper it seemed that we were only chasing optimal results rather than focusing upon the motivation behind this framework and how it leverages graph-ML to target a combinatorial problem on graphs. Thus, now in the updated version we focus on explaining the working of GBS more properly. First, the GCN based MIS Extraction which forms the basis of our algorithm is explained in detail. Then we focus upon explaining the use of beam search and metrics that are used to guide the process. We also updated the value-aware GNN section where we now clearly explain how the model is trained to accurately output MIS for reduced graph instances.
>
> (2) **Comment:** "... include truly state-of-the-art algorithms, and show that the proposed framework can outperform them....""
>
> **Revision:** Since, this is an ML-based model we were primarily interested in testing its performance against the other ML-based models introduced so far in the literature. However, we do agree that we must have included state-of-the-art non-ML based methods as well such as the metaheuristics based algorithms. We now compare GBS with the most recent metaheuristics  and report the results in table 2. GBS is able to produce near-optimal colorings which uses slightly more number of colors than the best known results.
>
> Also, one must note that the GBS method and these metaheuristics follows two very different approaches for graph coloring. The metaheuristics based solvers primarily begin with an initial k-coloring and then iteratively try to reduce color conflicts. Thus, they test for different values of k and report the minimum k required to attain conflict free coloring. Alternatively, the GBS method constructs a conflict-free coloring by sequentially identifying independent sets and assigning each set to a distinct color class. The coloring obtained by GBS comes with an additional property that is for coloring $C$ with color classes $(C_1,C_2,\ldots,C_k)$ we have $|C_1|\ge|C_2|\ge|C_3|\ldots\ge|C_k|$. This characteristic enhances the practical utility of GBS and makes it more suitable compared to other graph coloring solvers for a lot of applications such as register allocation, quantum circuit depth optimization etc. Thus, although our solver may not always attain the best solutions compared to existing methods, it can nevertheless be useful in a variety of real-world settings.

---

> ### Author Response · Authors · 2026-06-16
> **Response to Reviewer F9zs : Part 2**
>
> (3) **Comment:** "... expand the experimentation with Quantum Circuit Depth Optimization (QCDO), include additional details, and demonstrate the relevance of the results obtained for the literature on QCDO."
>
> **Revision:** The main aim of including QCDO in paper was to demonstrate how the proposed GBS framework can be applied to solve real-world graph-structured optimization problems. Primarily, we simply try to optimize circuit depth by finding a suitable ordering for gates so that large number of gates can be executed in parallel. We test our method on reversible arithmetic circuits such as adders, multipliers, and square-root implementations and randomly generated Clifford circuits. These circuits are chosen because they typically contain dense networks of CNOT and Toffoli gates. Although these networks give rise to complex dependency structures, these circuits often contain commuting gate subsets that can be exploited to increase parallelism.  As qiskit built-in function to optimize circuits i.e. Qiskit transpiler does not work as a gate ordering solver, we test our method against Qiskit DAG method. Through these experiments we can conclude that GBS can help in constructing a simple gate-reordering method that can efficiently explore the hidden parallelism and reduce circuit depth without altering the gate set or functionality.
>
> Thus, in the revised version we have tried to address all the weak points.  We have tried to improve paper presentation and include all the necessary references. Also, we have tried to improve experimental study to support our claims.  For Quantum Circuit Depth Optimization, we have tried to add more details regarding experiments and also discussed its practical utility for quantum computation.

---

### Review · Reviewer_WNgw · 2026-06-03

**Summary Of Contributions:**

The paper tackles the classic problem of Graph Coloring using a combination of GNN solvers + beam search techniques. The key contribution is an algorithm, Guided Beam Search (GBS), that maintains multiple partial colorings as state and explores residual graph states using heuristic metrics such as edge density, color efficiency, closeness centrality etc. It also extends the approach to a weighted objective of these metrics - the weighted GBS algorithm.

Further, the paper evaluates the approaches against classic baselines, on DIMACs graph datasets, and social network graph datasets etc.

**Audience:**

Yes

**Audience Explanation:**

It is a classic problem and it is being solved using GNNs + beam search. Some of these may be interesting to the community at large.

**Claims And Evidence:**

No

**Claims Explanation:**

I think the paper lacks some key baselines, which must be included before accepting. One obvious baseline to beat is to repeatedly apply Li et al. algorithm and color the obtained independent set (this is done using the classic algorithm but must be done with Li et al. as well).

Also, you must perform several ablations - for instances just Li et al. + beam search without the metrics;

**Requested Changes:**

As mentioned, include the missing benchmarks.

---

> ### Author Response · Authors · 2026-06-16
> **Response to Reviewer WNgw**
>
> We sincerely thank the reviewer for the careful reading of the manuscript and for the constructive suggestions. Below we address the comment made and summarize the corresponding revision.
>
> **Comment:** "One obvious baseline to beat is to repeatedly apply Li et al. algorithm and color the obtained independent set (this is done using the classic algorithm but must be done with Li et al. as well). Also, you must perform several ablations - for instances just Li et al. + beam search without the metrics."
>
> **Revision:** We agree that such baselines must be included as these will highlight the need of Guided Beam Search approach and the role of metrics in suitable MIS selection. In table 4, we now compare results obtained by different metrics against these baselines. We report average colors used and standard deviation obtained across 10 independent runs for all the methods. First baseline MIS+ Beam search without metric is referred to as Random. In this method at each step, out of the multiple MIS generated (all of same size), in the absence of metrics, we randomly choose c MIS where c is beam width.  Another baseline is repeatedly applying Li et al. MIS extraction algorithm  and coloring the obtained independent set. This is referred to as MIS-Color. The results indicate that metric guided beam search performs better than the baselines in terms of solution quality and is also more robust.